# DRIFT: Learning from Abundant User Dissatisfaction in Real-World Preference Learning

**Yifan Wang[1], Bolian Li[1], Junlin Wu[2], Zhaoxuan Tan[3], Zheli Liu[4], Ruqi Zhang[1],
Ananth Grama[1], Qingkai Zeng[4]\***

[1]Department of Computer Science, Purdue University
[2]Department of Computer Science and Engineering, Washington University in St. Louis
[3]Department of Computer Science and Engineering, University of Notre Dame
[4]College of Computer Science, Nankai University

`{wang5617, li4468, ruqiz, ayg}@purdue.edu, junlin.wu@wustl.edu`
`ztan3@nd.edu, liuzheli@nankai.edu.cn, qzengnkcs@gmail.com`

## Abstract

Real-world large language model deployments (e.g., conversational AI systems, code generation assistants) naturally generate abundant implicit user dissatisfaction (DSAT) signals, as users iterate toward better answers through refinements, corrections, and expressed preferences, while explicit satisfaction (SAT) feedback is scarce. Existing preference learning approaches are poorly aligned with this data profile, as they rely on costly human annotations or assume plentiful positive responses. In this paper, we introduce **DRIFT** (**D**issatisfaction-**R**efined **I**terative pre**F**erence **T**raining), which anchors training on real-world DSAT signals and samples positives dynamically from the evolving policy. Empirically, DRIFT models trained on real-world *WildFeedback* datasets and synthetic *UltraFeedback* datasets achieve up to +6.23% (7B) / +7.61% (14B) on WildBench Task Score and up to +8.95% (7B) / +12.29% (14B) on AlpacaEval2 win rate over base models, outperforming strong baseline methods such as iterative DPO and SPIN. At larger scales, the improvements are particularly pronounced: 14B models trained with DRIFT surpass GPT-4o-mini on WildBench. Further analysis shows that DRIFT also preserves exploratory capacity, yielding more diverse high-reward solutions rather than collapsing to narrow subsets. Theoretically, we demonstrate that this design preserves preference margins and avoids the gradient degeneration. These results show that DRIFT is an effective and scalable recipe for real-world post-training that leverages the most abundant and informative signal. Code and data are available at `https://github.com/cacayaya/DRIFT.git`.

## 1 Introduction

Large language models (LLMs) now power a wide range of real-world applications, including conversational assistants (e.g., GPT, Claude, Gemini), customer support, search and recommendation, productivity and education tools, and code generation. A key driver of this success is preference learning, a critical component of post-training that aligns model behavior with human judgments and values. Reinforcement Learning from Human Feedback (RLHF) (Ouyang et al., 2022) pioneered this approach by training a reward model on human preference data and subsequently optimizing the policy using reinforcement learning algorithms (Schulman et al., 2017). Direct Preference Optimization (DPO) (Rafailov et al., 2023) simplified this process by directly optimizing on preference pairs without requiring an explicit reward model, making the training procedure more stable and computationally efficient, while achieving comparable alignment performance.

However, these approaches depend on costly, carefully curated human preference annotations that are difficult to scale across domains and evolving user needs.In contrast, deployed LLM systems continuously generate vast amounts of real-world interaction data. Beyond offering scalability, such

---

*\*Corresponding author*

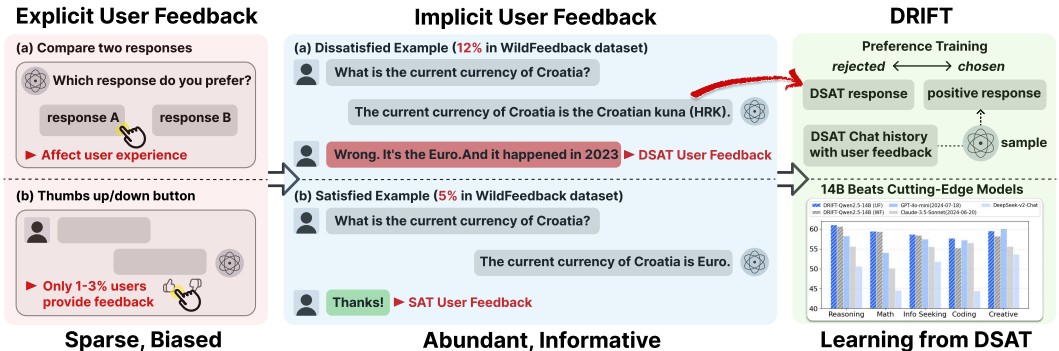

Figure 1: Overview of user feedback signals and the DRIFT framework. Explicit feedback (left) is sparse and biased, as most users are passive consumers. In contrast, implicit feedback (middle) provides abundant and informative signals, where dissatisfaction (DSAT) is far more prevalent than satisfaction (SAT) (e.g., 12% vs 5% in the *WildFeedback* dataset). DRIFT (right) leverages these DSAT signals for preference learning, enabling our 14B model to surpass commercial models.

real data often captures a richer and more nuanced spectrum of human preferences than small, curated annotation datasets, as users naturally convey satisfaction, dissatisfaction, and refinement intents during conversation. This motivates a key question:

> *How can we transform abundant but implicit user feedback from real-world interactions into scalable and effective preference learning signals for LLMs?*

From a data collection perspective, existing chatbot platform as shown in Figure 1 (left) attempts to gather user feedback through explicit mechanisms: (1) asking users to compare and rank multiple model responses, or (2) providing simple feedback buttons (e.g. thumbs up/down) at the end of chat interfaces. However, these collection methods are inefficient, as most are passive consumers (Lounamaa, 2024), with only 1–3% users willing to provide explicit feedback. Moreover, those who do provide feedback often express extreme opinions (strongly positive or negative) that may not reflect the broader distribution of user preferences. However, as illustrated by the example in Figure 1, (middle) users naturally express their preferences through the conversation itself through follow-up questions, correction requests, and iterative refinements, creating a rich source of implicit feedback. Beyond scalability, such real interaction data can contain richer and more representative preference information than curated annotation datasets, capturing fine-grained user intents that explicit labels often miss. Recent datasets such as WildChat-1M (Zhao et al., 2024) and LMSYS-Chat-1M (Zheng et al., 2024) have collected over one million real-world conversations, creating a rich foundation for studying naturally occurring user feedback. Building on these resources, several studies have explored ways to extract preference signals directly from user interactions. For example, Don-Yehiya et al. (2025) demonstrate that naturally occurring user feedback appears in approximately 30% of conversations and propose mining preferences by detecting explicit evaluative user responses. Similarly, *WildFeedback* (Shi et al., 2025) applies user satisfaction estimation (Lin et al., 2024b) to automatically extract satisfaction and dissatisfaction labels to construct large-scale preference datasets.

From a preference learning method perspective, recent works explored self-generated strategies to reduce reliance on human annotation. Self-Rewarding Language Models (Yuan et al., 2024) prompt the training model itself to score its own rollouts, but face a key limitation: the synchronized improvement of *chosen* and *rejected* responses progressively reduces their contrast, which in turn undermines effective preference learning (Wang et al., 2025a). An alternative approach, SPIN (Chen et al., 2024c) treats ground-truth responses from the SFT dataset as *chosen* and self-generated ones as *rejected*, yet is difficult to apply in practice where gold-standard responses are often rare, limiting its ability to generalize to broader scenarios. In contrast to positive feedback, dissatisfaction signals are naturally abundant as users refine suboptimal outputs through interaction. To leverage this underutilized signal, we introduce **DRIFT** (**D**issatisfaction-**R**efined **I**terative pre**F**erence **T**raining), a simple yet scalable method that directly leverages user dissatisfaction (DSAT) signals from authentic conversations to iteratively enhance model performance. Unlike SPIN, which fixes supervised responses as positives and treats self-generated ones as negatives, DRIFT anchors each training

pair with a real DSAT negative and samples the *chosen* responses from the current policy, enabling dynamic and policy-aligned adaptation. Our contributions are:

- **Empirical Validation**: DRIFT consistently surpasses other iterative self-improving methods, SPIN and IterDPO, yielding gains of up to +6.23% (7B) / +7.61% (14B) in WildBench Task Score and up to +8.95% (7B) / +12.29% (14B) in AlpacaEval2 win rate over base models.
- **Enhanced Exploration**: DRIFT preserves a larger exploration space and generates more diverse responses with substantially broader coverage of the global high-reward region.
- **Theoretical Analysis**: We show that DRIFT maintains a non-vanishing expected preference margin and prevents gradient collapse, which is a critical limitation in existing self-improving models.

## 2  RELATED WORK

### 2.1  LEARNING FROM REAL-WORLD USER FEEDBACK

Since relying on human labeling is not only expensive and time-consuming, but also highly subjective to a small set of annotators, recent work has shifted toward leveraging naturally occurring signals "in the wild". A natural starting point is to incorporate the most explicit feedback of user input, including edits and demonstrations. Gao et al. (2024) use edits in writing assistant settings to infer latent preferences, keeping the base LLM frozen and training a separate preference module that conditions future outputs. Similarly, Shaikh et al. (2025); Tucker et al. (2024) rely on a handful of user-provided demonstrations to bootstrap alignment, iteratively generating comparison pairs by treating user examples as preferred over LLM outputs and their intermediate revisions. Another stream of work draws on implicit feedback that emerges naturally during conversations. Hancock et al. (2019) introduced a self-feeding chatbot that monitors user satisfaction during deployment: satisfied turns are added as new training data, while explicit feedback is requested when dissatisfaction is detected. Liu et al. (2025) extended this idea, regenerating improved responses for dissatisfaction and applying them in supervised fine-tuning (SFT). While this provides some benefit on short tasks like MT-Bench (Bai et al., 2024), gains are limited on more complex real-world task benchmarks such as WildBench (Lin et al., 2024a). Building further on implicit signals, recent approaches transform them into pairwise preferences for direct optimization. Shi et al. (2025) identify dissatisfaction with GPT-4, summarize user preferences, and generate improved responses as *chosen* answers, contrasting them with the original unsatisfactory replies as *rejected*. Tan et al. (2025) follow a similar philosophy by extracting reader-centric questions from user-generated content, sampling multiple candidate answers with an LLM, and ranking them with a reward model to construct chosen–rejected pairs. In contrast, our approach requires no positive responses from stronger models, no reward model, and certainly no human-provided golden truth, relying solely on abundant real-world dissatisfaction (DSAT) signals and dynamic positives from the evolving policy.

### 2.2  SELF-IMPROVEMENT AND ITERATIVE DIRECT PREFERENCE OPTIMIZATION

Self-improvement strategies have emerged as an important avenue for iteratively enhancing model performance. SPIN (Chen et al., 2024c) formulates this framework by treating the previous iteration model as the opponent and the current iteration model as the main player, constructing preference data with the SFT response as the *chosen* response and the prior iteration's response as the *rejected* response, thereby fully utilizing the SFT data without requiring additional human annotation. However, pairing fixed ground-truth positives with self-generated negatives has been shown to risk reward hacking and model collapse (Wang et al., 2025b). Beyond this, Iterative DPO (Xiong et al., 2024; Xu et al., 2024) and Self-Rewarding Language Models (Yuan et al., 2024) and its variants (Pang et al., 2024; Chen et al., 2024a; Zeng et al., 2025; Tu et al., 2025; Chen et al., 2024b) explore generating on policy preference data via ranking responses by the model itself or a reward model / verifier and then conducting iterative DPO training. However, subsequent studies reveal that self-improving models face a critical limitation: the chosen and rejected responses can become too similar, leading to weak preference signals. To address this, Temporal Self-Rewarding LMs (Wang et al., 2025a) decouple chosen and rejected responses through past-future anchoring, while CREAM (Wang et al., 2025c) introduces consistency regularization to stabilize the preference signal. Our method naturally avoids this issue by anchoring on genuine DSAT negatives and sampling fresh positives from the evolving policy, thereby maintaining a non-vanishing preference margin and preventing gradient collapse.

## 3 DRIFT: DISSATISFACTION-REFINED ITERATIVE PRE-FERENCE TRAINING

User feedback in real-world systems is inherently asymmetric, while satisfied users rarely provided explicit positive responses, dissatisfied users are more likely to offer abundant and detailed feedback in the form of complaints, corrections, and stated preferences. As a result, dissatisfaction (DSAT) signals are not only more frequent but also richer in information than satisfaction (SAT) signals. Instead of viewing this imbalance as a limitation, DRIFT exploits it by treating authentic dissatisfaction as high-quality negative supervision, while generating positive feedback dynamically from the evolving model itself.

Our approach is motivated by two key insights:

- **Genuine dissatisfaction reflects real deployment failure modes**, offering more informative and reliable supervision than synthetically constructed negatives.

- **Iteratively sampling fresh positives from the current policy** maintains the margin between chosen and rejected responses, thus mitigating the gradient collapse that plagues most self-improvement methods as these responses become increasingly similar over time.

Formally, let $\mathcal{X}$ denote the prompt space and $\mathcal{Y}$ the response space. The current model is $\pi_\theta : \mathcal{X} \times \mathcal{Y} \to (0, 1)$, and $\pi_{\mathrm{ref}}$ is a frozen reference model. Let $\mathcal{X}_{\mathrm{DSAT}} \subseteq \mathcal{X}$ denote prompts with observed dissatisfaction signals. For each $x \in \mathcal{X}_{\mathrm{DSAT}}$, we observe a set of negative responses:

$$\mathrm{DSAT}(x) = \{\, y^- : \text{user expressed dissatisfaction} \}. \tag{1}$$

DRIFT proceeds in iterative refinement cycles, where each round builds upon the improved policy from the previous iteration (Algorithm 1). We begin by filtering the wild dataset to extract dissatisfaction (DSAT) cases, producing prompt-response pairs $(x, y^-)$ that reflect concrete failure modes encountered in real-world scenarios. At each iteration, the current model $\pi_{\theta_k}$ generates a fresh positive response $y^+$ for the same prompt $x$, allowing the positive response to evolve alongside the model's capacities. The model is then updated by minimizing the DPO loss:

$$\mathcal{L}_{\mathrm{DPO}} = -\mathbb{E}_{(x,y^+,y^-)} \left[ \log \sigma \left( \beta \log \frac{\pi_\theta(y^+|x)}{\pi_{\mathrm{ref}}(y^+|x)} - \beta \log \frac{\pi_\theta(y^-|x)}{\pi_{\mathrm{ref}}(y^-|x)} \right) \right] \tag{2}$$

where $\beta$ controls the preference margin and $\sigma$ denotes the logistic function.

---

**Algorithm 1** DRIFT: Dissatisfaction-Refined Iterative Preference Training

---

1: **Input:** Wild implicit feedback dataset, current model $\pi_\theta$, reference model $\pi_{\mathrm{ref}}$, number of iterations $K$
2: **Output:** Updated model parameters $\theta_K$
3: **Filter:** Extract DSAT signals to form $\mathcal{D} = \{(x, y^-) \mid y^- \in \mathrm{DSAT}(x)\}$
4: **for** $k = 1, \ldots, K$ **do**
5:     **Positive Sampling:** For each $(x, y^-) \in \mathcal{D}$, sample a fresh positive response $y^+ \sim \pi_{\theta_k}(\cdot \mid x)$
6:     **Loss Update:** Update $\theta_k$ by minimizing $\mathcal{L}_{\mathrm{DPO}}$ (Eq. 2)
7: **end for**

---

## 4 EXPERIMENT

In this section, we evaluate DRIFT against strong self-improvement baselines, focusing on real world task performance. Sec. 4.1 outlines datasets, training recipe, and evaluation benchmarks. Sec. 4.2 presents the task performance on WildBench and AlpacaEval2. Then, in Sec. 4.3, we analyze exploration cability on response space of each method through global high-reward coverage.

### 4.1 SETUP

**Datasets.** *WildFeedback* **(real-world, user-feedback).** The *WildFeedback* dataset is derived from WildChat-1M, a corpus of over one million human–ChatGPT conversations, by assigning per-turn

labels Satisfaction (SAT), Dissatisfaction (DSAT), Neutral (Non-DSAT/Non-SAT). Labels are derived using SPUR (Lin et al., 2024b), which recursively prompts GPT-4 to learn SAT/DSAT rubrics from thumb-annotated conversations and applies them to score satisfaction/ dissatisfaction. Among all 88,920 unique conversations, only 4,478 (5.04%) conversations were labeled SAT, while 10,632 (11.96%) were labeled DSAT, which is more than twice the SAT count. We also curate 491 seed data items (0.55%) in which LLM responses transition from DSAT to SAT after revision, naturally yielding preference pairs.

***UltraFeedback* (synthetic, LLM-labeled).** For completeness and comparability with prior self-improvement work, we also evaluate on *UltraFeedback* in which each prompt has four completions from different models that are scored by GPT-4. This synthetic setting provides a complementary evaluation to the real-world data setting and ensures fair comparison with SPIN/ IterDPO on commonly used LLM-labeled preference data.

Table 1: Comparison of preference data construction strategies across different methods. "Self-Gen" means responses are generated by the current policy. "Real" is from user feedback.

| Method | Chosen Response | | Rejected Response | | No Positive Examples | Leverage Real User Feedback |
|--------|-----------------|------|-------------------|------|------------|-------------|
| | Self-Gen | Real | Self-Gen | Real | | |
| SPIN | ✗ | ✓ (SAT) | ✓ | ✗ | ✗ | ✓ |
| IterDPO | ✓ | ✗ | ✓ | ✗ | ✓ | ✗ |
| DRIFT (Ours) | ✓ | ✗ | ✗ | ✓ (DSAT) | ✓ | ✓ |

**Training Recipe.** Our experiments are conducted on Qwen2.5-7B-Instruct and Qwen2.5-14B-Instruct. We adopt a two-stage training:
(1) *Warm start:* train on the 491 seed DSAT→SAT pairs, which provides an initial aligned policy.
(2) *Iterative preference training:* After warm start, each method constructs fresh preference pairs.
*Per-iteration preference data construction (Table 1):*
DRIFT: In *WildFeedback*, we keep the DSAT reply as the *rejected* response and, at each iteration, sample a fresh response from the current policy using prompt which contains the full conversation including the DSAT user turn and an explicit improvement instruction. In *UltraFeedback*, we replace the original *chosen* with a fresh policy sample.
SPIN: In *WildFeedback*, we use the SAT reply as the *chosen* response and a policy sample as the *rejected* response using the prompt which is the conversation before the SAT user turn. In *UltraFeedback*, we replace the original *rejected* with a fresh policy sample.
IterDPO: In *WildFeedback*, we generate two responses using different prompts: the *chosen* context includes the full conversation including the DSAT user turn and an explicit improvement instruction, while the *rejected* context is the conversation before the DSAT user turn which does not reveal the user preference and instruction information. In *UltraFeedback*, both responses are generated from the same prompt and ranked by the reward model [1]; the higher-scored response is *chosen* and the other is *rejected*.
We then perform one epoch of DPO training after data generation, which prevents overfitting during iterative training. Full training details are presented in Appendix D.

**Evaluation.** We evaluate on WildBench (Elo, Task Score) and AlpacaEval2 (win rate, length-controlled; LC). WildBench is built from challenging real-world user queries in WildChat-1M and spans five diverse categories: *Creative*, *Reasoning*, *Math*, *Info Seek*, and *Coding*, making it well suited for assessing our method for real-world performance.

## 4.2 PERFORMANCE EVALUATION

### 4.2.1 RESULTS ON WILDFEEDBACK

We first examine performance on the real-world *WildFeedback* dataset, which contains authentic user satisfaction/ dissatisfaction labels and exhibits a strong imbalance: dissatisfied responses (DSAT) outnumber satisfied ones (SAT) by more than 2:1. Hence, we consider two configurations: a **Controlled setting** with around 4k samples (matching SPIN for fair comparison), and a **Full setting** using all 11k DSAT samples to demonstrate DRIFT's ability to exploit abundant negative feedback.

---

[1]OpenAssistant/reward-model-deberta-v3-large-v2

Table 2: Results of training on *WildFeedback*. Appendix C for detailed per-task results.

| Method | WildBench | | | | AlpacaEval2 | | | |
| | 7B | | 14B | | 7B | | 14B | |
| | Elo | Score | Elo | Score | Win | LC | Win | LC |
|---|---|---|---|---|---|---|---|---|
| Base | 1194.67 | 48.66 | 1213.17 | 55.08 | 37.69 | 39.73 | 36.65 | 43.58 |
| Seed | 1193.66 | 49.11 | 1213.50 | 54.93 | 42.67 | 42.06 | 40.25 | 45.78 |
| **Controlled Setting** | | | | | | | | |
| *SPIN* | | | | | | | | |
| iter1 | 1180.75 | 42.86 | 1200.63 | 47.16 | 26.21 | 34.09 | 25.53 | 37.28 |
| iter2 | 1173.45 | 37.86 | 1192.56 | 44.04 | 20.56 | 29.00 | 18.57 | 30.91 |
| *IterDPO* | | | | | | | | |
| iter1 | 1189.43 | 47.07 | 1206.65 | 51.79 | 41.55 | 41.35 | 37.14 | 43.14 |
| iter2 | 1192.46 | 48.94 | 1211.69 | 56.63 | 41.18 | 40.14 | 48.32 | **47.28** |
| *DRIFT (Ours)* | | | | | | | | |
| iter1 | **1197.13** | **51.06** | **1215.73** | **58.37** | 42.73 | 41.41 | **48.76** | 45.42 |
| iter2 | 1195.33 | **51.06** | 1214.03 | 57.59 | **43.79** | **41.49** | 46.83 | 43.48 |
| **Full Setting** | | | | | | | | |
| *IterDPO* | | | | | | | | |
| iter1 | 1185.11 | 46.31 | 1205.48 | 52.34 | 40.36 | 39.85 | 38.07 | 43.99 |
| iter2 | 1182.33 | 46.17 | 1206.63 | 51.38 | 35.76 | 37.06 | 32.88 | 37.92 |
| *DRIFT (Ours)* | | | | | | | | |
| iter1 | 1194.81 | 50.61 | 1212.83 | 57.27 | 43.90 | 40.32 | **48.63** | **47.46** |
| iter2 | **1199.09** | **51.69** | **1217.61** | **58.30** | **46.64** | **42.72** | 45.33 | 44.93 |

As shown in Table 2, DRIFT raises the WildBench Task Score by 6.23% (+3.03) for 7B and 5.97% (+3.29) for 14B, and boosts AlpacaEval2 win rate by 8.95% for 7B and 12.11% for 14B compared to the base models. And our method consistently outperforms both SPIN and IterDPO across all metrics in both controlled and full data settings.

While SPIN shows degraded performance with iterations likely due to its reliance on a fixed set of satisfied responses becoming stale, DRIFT maintains steady improvements, suggesting that its strategy prevents distribution shift. IterDPO performs better than SPIN but still lags behind DRIFT in both settings, indicating that while reward model guidance helps, the real world informative DSAT examples provides superior training signal. Notably, DRIFT's controlled setting (using only 4k samples) already matches or exceeds IterDPO's full setting performance, demonstrating the efficiency of dissatisfaction-anchored learning. The stronger gains at the 14B scale suggest that DRIFT benefits larger models more, likely because their greater capacity makes it easier to discover better positives while being anchored by real negatives. This making DRIFT well suited for scaling up.

**Long-horizon stability beyond 2 iterations.** To further investigate the stability and performance trends for longer iterations, we extended all methods to five iterations on Qwen2.5-7B. Figure 2 visualizes the performance trajectories across iterations.

Both SPIN and IterDPO peak early at iter1 and then exhibit performance degradation, with SPIN showing the most pronounced decline. In contrast, DRIFT demonstrates sustained improvement up to iter4 (52.47), and then forms a stable plateau with minimal variation (51.22 at iter5). This stability suggests that DRIFT's

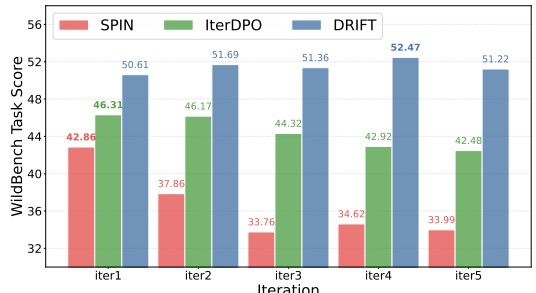

Figure 2: Performance across five iterations on Qwen2.5-7B. Bold values indicate the maximum score achieved by each method across iterations.

strategy of anchoring on real dissatisfied responses while continuously sampling fresh positives prevents mode collapse that plagues other self-play or self-improve methods during extended iterative training. The limited performance collapse observed even at iter5 further validates DRIFT's robustness for long-horizon self-improvement.

**Unguided ablation study and the source of DRIFT's advantage.** An important consideration is whether DRIFT's observed gains are attributable to the guidance prompts provided during positive generation. To examine this, we evaluated an unguided variant in which DRIFT and IterDPO generate responses solely from the original user prompt, matching SPIN's configuration. The results (see Appendix C, Table 5) show that DRIFT remains consistently stronger than both baselines in this setting, and the gap between the unguided and full versions is small. This suggests that DRIFT's gains are not attributable to instruction prompt and reference point, but instead to its design principle of anchoring on off-policy DSAT negatives while drawing positives from the evolving policy.

### 4.2.2 RESULTS ON ULTRAFEEDBACK

To ensure comprehensive evaluation and fair comparison with prior work, we also evaluate on the synthetic *UltraFeedback* dataset, where preferences are scored by GPT-4 rather than derived from real user interactions. This complementary evaluation helps assess whether DRIFT's advantages generalize beyond the specific characteristics of real-world dissatisfaction signals to more conventional preference learning settings. As shown in Table 3, DRIFT outperforms the base model with gains of 4.62% (+2.25) for 7B and 7.61% (+4.19) for 14B on WildBench Task Score, and improvements of +3.35% (7B) and +12.29% (14B) on AlpacaEval2 win rate. Compared to the best SPIN/IterDPO results, DRIFT achieves additional gains of +2.14 (7B) and +4.49 (14B) on Task Score, as well as +6.51% (7B) and +6.10% (14B) on win rate.

Table 3: Results of training on *UltraFeedback*. Appendix C for detailed per-task results.

| Method | WildBench | | | | AlpacaEval2 | | | |
| --- | --- | --- | --- | --- | --- | --- | --- | --- |
| | 7B | | 14B | | 7B | | 14B | |
| | Elo | Score | Elo | Score | Win | LC | Win | LC |
| Base | 1194.67 | 48.66 | 1213.17 | 55.08 | 37.69 | 39.73 | 36.65 | 43.58 |
| *SPIN* | | | | | | | | |
| iter1 | 1163.16 | 35.10 | 1178.88 | 36.99 | 18.39 | 26.62 | 16.09 | 28.20 |
| iter2 | 1139.66 | 25.93 | 1155.07 | 28.01 | 13.23 | 19.91 | 13.66 | 24.29 |
| *IterDPO* | | | | | | | | |
| iter1 | 1194.45 | 48.77 | 1214.14 | 54.21 | 34.53 | **40.55** | 33.29 | 42.84 |
| iter2 | 1192.01 | 48.49 | 1215.12 | 54.78 | 32.15 | 39.92 | 28.51 | 40.47 |
| *DRIFT (Ours)* | | | | | | | | |
| iter1 | 1197.04 | **50.91** | 1215.67 | 58.52 | **41.04** | 40.37 | 47.89 | **48.46** |
| iter2 | **1197.94** | 50.32 | **1218.75** | **59.27** | 40.47 | 37.09 | **48.94** | 47.43 |

### 4.3 EXPLORATORY CAPACITY ANALYSIS: DRIFT EXPLORES MORE DIVERSE HIGH REWARD SOLUTIONS

A central question in preference learning is whether pushing rewards upward narrows the response distribution and erodes *exploration capacity* and *diversity*. Methods that optimize aggressively for top scores or fixed chosen responses can shift toward mode seeking: peak metrics improve while alternative high-quality modes are under explored. We further investigate whether DRIFT's strategy of sampling fresh positives while anchoring on real dissatisfied responses enhances the model's ability to explore the space of high-quality solutions than SPIN or iterDPO, which may progressively constrain the solution space.

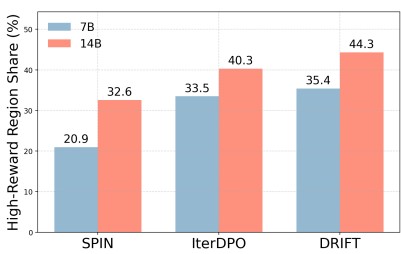

Figure 3: Comparison of high reward region coverage.

**Semantic reward topography construction.** For each prompt, we first sample 128 responses from each method and embed all collected responses[2]. We obtain a 2D semantic projection via UMAP and estimate a reward-weighted density surface $Z_{\text{all}}(g) \in [0, 1]$ over a regular grid $g$ using Gaussian KDE. The global high-reward region is defined as

$$\mathcal{H} = \{ g : Z_{\text{all}}(g) \geq z_{\text{high}} \}, \qquad z_{\text{high}} = \text{Quantile}(Z_{\text{all}}, 0.8).$$

For each method $m$, we construct a corresponding surface $D_m(g)$ on the same grid and bandwidth, and measure its coverage inside the global high-reward region by

$$\mathcal{S}_m = \{ g \in \mathcal{H} : D_m(g) \geq z_{\text{high}} \}, \qquad \text{Share}(m) = \frac{|\mathcal{S}_m|}{|\mathcal{H}|}.$$

We render $Z_{\text{all}}$ as the background terrain, overlay the boundary of $\mathcal{H}$ (dashed), and shade $\mathcal{S}_m$ for each method. We report the average high-reward coverage across 50 prompts in Figure 3. DRIFT consistently attains the largest share at both 7B and 14B scales, with a larger margin at 14B, indicating greater high reward solutions diversity and better scalability.

**Case Study: High-Reward Coverage and Response Diversity.** Figure 4 shows an example that DRIFT distributes responses across a broader set of high-reward semantic regions, whereas SPIN and IterDPO concentrate their outputs within a much smaller subset of the space. Notably, DRIFT also discovered a distinct region (circled in the reward topography) where it uniquely employed markdown formatting to structure research papers, demonstrating alternative presentation styles for the same prompt.

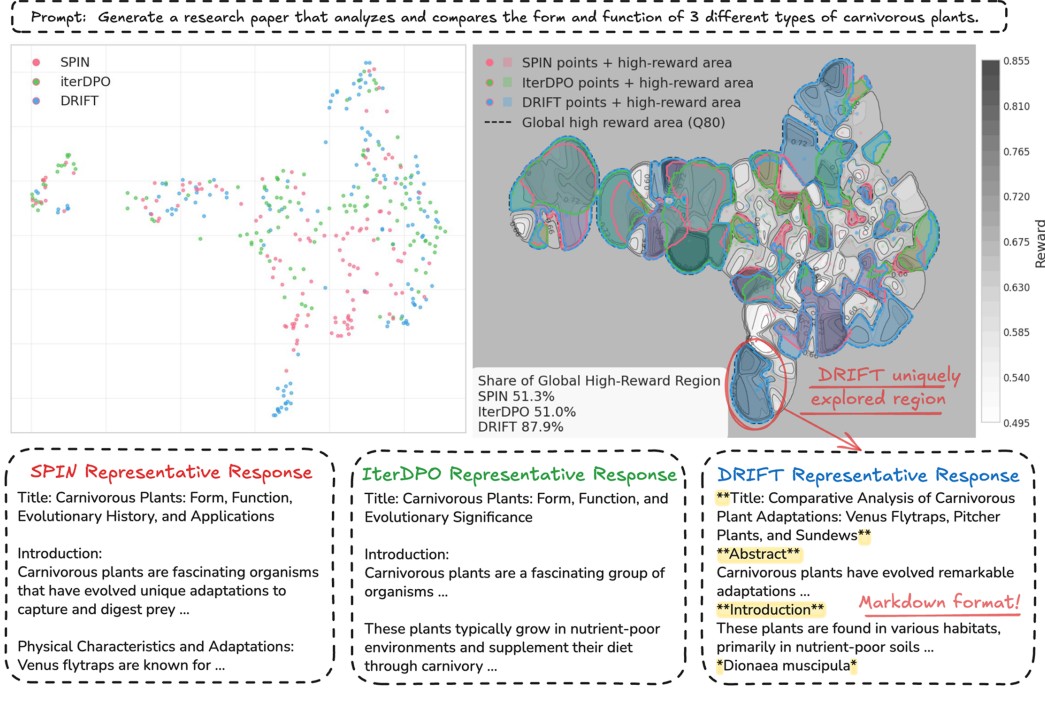

Figure 4: Example of response diversity and quality comparison via semantic reward topography. Two central plots: Left is the UMAP scatter of all responses; Right is the semantic reward topography showing the global high-reward region and the coverage of the three methods. Full prompt and responses are in Appendix E

Anchoring on authentic DSAT negatives while sampling fresh positives enables DRIFT to maintain and amplify exploratory capacity. DRIFT not only reaches high reward but also occupies a far broader set of high-reward areas, explaining why it continues to improve over iterations without collapsing to a small family of solutions.

---

[2]Embeddings are computed with `https://huggingface.co/Qwen/Qwen3-Embedding-0.6B`.

## 5  THEORETICAL ANALYSIS

DRIFT shows superior performance by leveraging real-world user dissatisfaction as high-quality negatives, which leads us to further investigate and analyze how real-world data shapes the success of DRIFT and why some other strong baselines like SPIN and IterDPO fall short. In this section, we prove that DRIFT maintains a usable gradient signal through fresh positives anchored by genuine DSAT negatives; in contrast, updates fitted to a fixed SAT set like SPIN can easily overfit to a small sub-optimal subset with gradient collapse.

**Notation.**  Let $\mathcal{X}$ be the prompt set. For a particular prompt $x \sim \mathcal{X}$, denote $y^+$ as the chosen response and $y^-$ the rejected response. Let the generation likelihoods of $y^+$ and $y^-$ to be $\pi^+ = \pi_\theta(y^+ \mid x)$ and $\pi^- = \pi_\theta(y^- \mid x)$ respectively, where $\pi_\theta$ is the language model we aim to train.

For formula simplicity, we denote the implicit reward margin (Rafailov et al., 2023) as:

$$s = \beta \cdot \left( \log \left( \pi^+ / \pi_{\text{ref}}(y^+ | x) \right) - \log \left( \pi^- / \pi_{\text{ref}}(y^- | x) \right) \right), \tag{3}$$

and the loss function:

$$\ell = - \log \sigma(s), \qquad \nabla_\theta \ell = - \beta \, \sigma(-s) \Big[ \nabla_\theta \log \pi_\theta(y^+ \mid x) - \nabla_\theta \log \pi_\theta(y^- \mid x) \Big], \tag{4}$$

where $\sigma(s) = (1 + e^{-s})^{-1}$ is the logistic function. We also denote $d_\theta := \nabla \ln \pi_\theta(y^+ \mid x) - \nabla \ln \pi_\theta(y^- \mid x)$ and $g(\theta) := \mathbb{E}[-\nabla \ell(\theta)] = \beta \, \mathbb{E}[\sigma(-s) \, d_\theta]$. Finally, let $r^\star : \mathcal{X} \times \mathcal{Y} \to [0, 1]$ be the unknown gold reward, and $J(\theta) := \mathbb{E}_{x \sim \mu} \, \mathbb{E}_{Y \sim \pi_\theta(\cdot | x)}[r^\star(x, Y)]$ be the overall objective.

**Reward Margin Hypothesis.**  With a probability of at least $p_{\text{imp}}$, both the implicit reward margin and the gold reward margin have positive lower bounds. Specifically, there exists some $\tau \in (0, \frac{1}{2}]$ and $m_r > 0$ such that

$$E_{\text{imp}} := \left\{ \sigma(-s) \geq \tau \text{ and } r^\star(x, y^+) - r^\star(x, y^-) \geq m_r \right\}, \qquad \mathbb{P}(E_{\text{imp}}) \geq p_{\text{imp}} > 0. \tag{5}$$

This hypothesis ensures that chosen responses are mostly ranked higher than rejected responses.

**Non-vanishing expected training signal.**  We first certify that DRIFT maintains a uniform *expected* gradient magnitude under a positive-mass "quality" event.

**Lemma 1** (Expected gradient lower bound under local quality). *Let $E = \{\sigma(-s) \geq \tau\}$ with $\mathbb{P}(E) \geq p_0 > 0$ for some $\tau \in (0, \frac{1}{2}]$. If $\mathbb{E}\big[\|d_\theta\| \mid E\big] \geq \Delta_{\text{cond}} > 0$, then*

$$\mathbb{E}\big[ \|\nabla_\theta \ell\| \big] \geq \beta \, \tau \, p_0 \, \Delta_{\text{cond}}. \tag{6}$$

Proof. *From Eq. 4 and $\sigma(-s) \geq 0$,*

$$\mathbb{E}\|\nabla \ell\| = \mathbb{E}\big[\beta \, \sigma(-s) \, \|d_\theta\|\big] \geq \beta \, \tau \, \mathbb{E}\big[\|d_\theta\| \mathbf{1}_E\big] = \beta \, \tau \, \mathbb{P}(E) \, \mathbb{E}\big[\|d_\theta\| \mid E\big].$$

This bound shows that as long as a non-negligible fraction of pairs satisfy $\sigma(-s) \geq \tau$ and have a nonzero conditional gradient gap $\big(\mathbb{E}[\|d_\theta\| \mid \sigma(-s) \geq \tau] > 0\big)$, the expected training signal stays away from zero.

**Expected improvement of actual utility.**  We now state a general improvement guarantee: the expected DPO step increases the true utility $J$, with the gain quantified based on the key data condition.

**Theorem 1** (Expected improvement of $J$). *Assume the improvement event Eq. 5 holds with probability at least $p_{\text{imp}}$, and there exists $\lambda > 0$ such that*

$$\mathbb{E}\Big[ \big\langle \nabla J(\theta), \, d_\theta \big\rangle \,\Big|\, E_{\text{imp}} \Big] \geq \lambda. \tag{7}$$

*If $J$ is $L_J$-smooth in a neighborhood of $\theta$, then for any $\eta > 0$,*

$$\mathbb{E}\big[J(\theta + \eta \, g(\theta))\big] \geq J(\theta) + \eta \, \beta \, \tau \, p_{\text{imp}} \, \lambda - \frac{L_J}{2} \eta^2 \, \mathbb{E}\big[\|g(\theta)\|^2\big]. \tag{8}$$

*In particular, for sufficiently small $\eta$, the right-hand side exceeds $J(\theta)$ by a linear-in-$\eta$ margin $\beta \tau p_{\text{imp}} \lambda$ up to $O(\eta^2)$.*

*Proof sketch.* $g(\theta) = \beta \, \mathbb{E}[\sigma(-s)d_\theta]$ gives

$$\mathbb{E}\langle \nabla J, g\rangle = \beta \, \mathbb{E}\big[\sigma(-s)\langle \nabla J, d_\theta\rangle\big] \geq \beta\,\tau\,\mathbb{P}(E_{\mathrm{imp}})\,\mathbb{E}\big[\langle \nabla J, d_\theta\rangle \mid E_{\mathrm{imp}}\big] \geq \beta\,\tau\,p_{\mathrm{imp}}\,\lambda.$$

$L_J$-smoothness yields $J(\theta + \eta g) \geq J(\theta) + \eta\langle \nabla J, g\rangle - \frac{L_J}{2}\eta^2\|g\|^2$, then take expectations to obtain Eq.8. *Full proof in Appendix B.2.* □

**Why DRIFT outperforms SPIN?** When SPIN concentrates probability on a finite SAT catalogue and reaches a fixed point, the magnitude of the pairwise DPO signal on SPIN pairs is controlled by the catalogue's log–density-ratio variance:

$$\left\|\mathbb{E}\big[\beta\,\sigma(-s)\,d_{\hat\theta}\big]\right\| \leq \frac{\beta}{4}\,\sqrt{\mathrm{Var}(s)}\,\sqrt{\mathbb{E}\|d_{\hat\theta}\|^2}, \qquad \mathrm{Var}(s) = 2\beta^2\,\mathrm{Var}_{Y\sim p_{\mathrm{SAT}}}\big(h(Y)\big), \quad (9)$$

where $h(y) := \ln\pi_{\hat\theta}(y \mid x) - \ln\pi_{\mathrm{ref}}(y \mid x)$ (Proposition 1 in Appendix). Thus, a small $\mathrm{Var}_{p_{\mathrm{SAT}}}(h)$ implies a weak training signal that can quantitatively *degenerate*. By contrast, DRIFT maintains a non-vanishing signal and practical gains. Full discussions are in Appendix B.3.

---

**Summary**

- **Signal***: If improvement events occur with nonzero probability, DRIFT's expected gradient stays non-vanishing (Lemma 1).
- **Utility***: Under the local correlation, a small step along the expected DPO direction improves $J$ up to $O(\eta^2)$ (Theorem 1).
- **Contrast***: At SPIN fixed points on a finite SAT catalogue, the signal is controlled by catalogue log–density-ratio variance and can *degenerate* (Proposition 1).

---

## 6 DISCUSSION

**Generalization.** To test our method's generality, we applied the same training procedure to Gemma-3-12B-it, a multimodal and structurally distinct architecture. The results (see Appendix C, Table 6) show that DRIFT again outperformed SPIN and IterDPO across all WildBench categories, exhibiting similar stability and improvement patterns. These results confirm that DRIFT generalizes beyond a single model family and performs well even on different model architecture.

**Safety implications of training with DSAT signals.** Since DRIFT explicitly anchors on real user dissatisfaction, it is important to assess whether such supervision inadvertently amplifies adversarial vulnerabilities or demographic biases. Evaluations on AdvBench and ToxiGen (see Appendix C, Table 7) show that DRIFT does not increase jailbreak success rates, toxicity, or group-specific harms relative to baseline models across iterations.

Taken together, these results show that DRIFT's asymmetric pairing of off-policy DSAT negatives with on-policy positives provides a stable and informative learning signal, generalizes across model families, and preserves baseline safety characteristics. Grounding preference optimization in real dissatisfaction thus offers a reliable and scalable direction for real-world post-training.

## 7 CONCLUSION

Real-world post-training rarely comes with abundant golden positives; it comes with abundant dissatisfaction and iterative user edits. In this paper, we introduced DRIFT, a simple, scalable recipe that pairs authentic DSAT negatives with policy sampled positives, turning in-situ feedback into stable, exploration preserving updates. Empirically, on real-world user feedback dataset *WildFeedback*, DRIFT outperforms SPIN and IterDPO on WildBench and AlpacaEval2 (with the stronger margins at larger base models); on synthetic LLM-labeled dataset *UltraFeedback*, it retains its superiority. Exploratory capacity analysis indicates that DRIFT explores more diverse high-reward solutions rather than overfitting to a narrow region. Theoretically, we show that DRIFT's admits a uniform, non-vanishing gradient lower bound, avoiding the collapse that arises when training concentrates probability on a finite fixed *chosen* (or SFT) set as in SPIN. Together, these results suggest DRIFT is a promising practical recipe for preference learning with real-world user feedback.

**Ethics Statement**    This work relies on a publicly datasets *WildFeedback* that contain anonymized human–LLM conversations. No personally identifiable information was collected or used. All experiments comply with dataset licenses and terms of use. The research has no foreseeable negative social or ethical impacts.

**Reproducibility Statement**    We have made extensive efforts to ensure the reproducibility of our work. The main paper details our training setup, datasets, and evaluation metrics (Secs. 4.1–4.3). Dataset construction and filtering steps are described in Sec. 4.1, with references to the source corpora. Training hyperparameters, iteration procedures and training dynamics are presented in Appendix D. To further facilitate verification and reuse, we will open-source our code, including data-processing pipelines, training scripts and analysis upon paper acceptance.

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

## A  DISCLOSURE OF LLM USE IN PAPER PREPARATION

We acknowledge the use of LLMs for assistance with writing and polishing text. All the content suggested by LLMs in writing was proofread and manually adjusted before being integrated into the final manuscript. The authors take full responsibility for the accuracy and factuality of all content presented.

## B  THEORETICAL PROOFS

### B.1  ASSUMPTIONS

(A1) **Improvement event (data-level).** There exist $\tau \in (0, \frac{1}{2}]$, $m_r > 0$, and $p_{\mathrm{imp}} > 0$ such that, with
$$E_{\mathrm{imp}} \;=\; \big\{\, \sigma(-s) \geq \tau \text{ and } r^\star(x, y^+) - r^\star(x, y^-) \geq m_r \,\big\},$$
one has $\mathbb{P}(E_{\mathrm{imp}}) \geq p_{\mathrm{imp}}$.

(A2) **Local smoothness of $J$.** There exists $L_J < \infty$ such that, for all sufficiently small $v$,
$$J(\theta + v) \;\geq\; J(\theta) + \langle \nabla J(\theta), v \rangle - \tfrac{L_J}{2}\, \|v\|^2.$$

(A3) **Finite second moment for the score difference.** $\mathbb{E}\, \|d_\theta\|^2 \;\leq\; C_d < \infty$.

### B.2  PROOF OF THEOREM 1 (EXPECTED IMPROVEMENT OF $J$))

**Theorem** (Restatement of Theorem 1). *Under Assumptions (A1), (A2), (A3), and the local advantage correlation condition Eq.7, for any $\eta > 0$ and $g(\theta) = \beta\, \mathbb{E}[\sigma(-s)\, d_\theta]$,*
$$\mathbb{E}\big[J(\theta + \eta\, g(\theta))\big] \;\geq\; J(\theta) \;+\; \eta\,\beta\,\tau\, p_{\mathrm{imp}}\,\lambda \;-\; \frac{L_J}{2}\, \eta^2\, \beta^2\, C_d.$$

*Proof.* By definition, $g(\theta) = \beta\, \mathbb{E}\big[\sigma(-s)\, d_\theta\big]$. Taking inner product with $\nabla J(\theta)$ and then expectation,
$$\mathbb{E}\, \langle \nabla J(\theta),\, g(\theta) \rangle = \beta\, \mathbb{E}\Big[\sigma(-s)\, \langle \nabla J(\theta), d_\theta \rangle\Big].$$
On the improvement event from Assumption (A1), $\sigma(-s) \geq \tau$; conditioning on $E_{\mathrm{imp}}$ and using Eq.7,
$$\mathbb{E}\, \langle \nabla J(\theta),\, g(\theta) \rangle \;\geq\; \beta\,\tau\, \mathbb{P}(E_{\mathrm{imp}})\, \mathbb{E}\big[\langle \nabla J(\theta), d_\theta \rangle \mid E_{\mathrm{imp}}\big] \;\geq\; \beta\,\tau\, p_{\mathrm{imp}}\, \lambda.$$
By the $L_J$-smoothness in Assumption (A2),
$$J(\theta + \eta g) \;\geq\; J(\theta) + \eta\, \langle \nabla J(\theta), g \rangle - \tfrac{L_J}{2}\, \eta^2\, \|g\|^2.$$
Taking expectations and combining the previous bound gives
$$\mathbb{E}\, J(\theta + \eta g) \;\geq\; J(\theta) \;+\; \eta\,\beta\,\tau\, p_{\mathrm{imp}}\,\lambda \;-\; \tfrac{L_J}{2}\, \eta^2\, \mathbb{E}\, \|g\|^2.$$
It remains to bound $\mathbb{E}\, \|g(\theta)\|^2$. Since $\sigma \in (0, 1)$,
$$\|g(\theta)\| = \big\|\beta\, \mathbb{E}[\sigma(-s)\, d_\theta]\big\| \leq \beta\, \mathbb{E}\|d_\theta\| \leq \beta\, \sqrt{\mathbb{E}\|d_\theta\|^2} \leq \beta\, \sqrt{C_d},$$
where we used Assumption (A3) and Jensen. Hence $\mathbb{E}\, \|g(\theta)\|^2 \leq \beta^2 C_d$, which yields the stated inequality. $\qquad\square$

### B.3  SPIN REAL-WORLD PERFORMANCE DISCUSSION

SPIN updates concentrate probability on the finite SAT catalogue $\mathrm{SAT}(x)$ and, under a trust-region style update, admit the closed-form iteration (see, e.g., (Chen et al., 2024c)):
$$p_{\theta_{t+1}}(y \mid x) \;\propto\; p_{\theta_t}(y \mid x) \Big[\frac{p_{\mathrm{SAT}}(y \mid x)}{p_{\theta_t}(y \mid x)}\Big]^{1/\lambda}, \tag{5.3}$$
which drives $p_{\theta_t}(\cdot \mid x)$ toward $p_{\mathrm{SAT}}(\cdot \mid x)$ supported on $\mathrm{SAT}(x)$. Consequently, at any fixed point $\hat\theta$ one has $\pi_{\hat\theta}(\cdot \mid x) = p_{\mathrm{SAT}}(\cdot \mid x)$; under the SPIN data rule (positive sampled from $p_{\mathrm{SAT}}$, negative sampled independently from $\pi_{\hat\theta}$ given $x$), the positive $y^+$ and negative $y^-$ are conditionally i.i.d. on $\mathrm{SAT}(x)$ with the common marginal $p_{\mathrm{SAT}}$. The DPO signal then depends on the *variance* of the log-density ratio on that finite set and can quantitatively degenerate.

**Proposition 1** (Quantitative degeneration at a SPIN fixed point). *At a SPIN fixed point $\hat{\theta}$ with $\pi_{\hat{\theta}}(\cdot|x) = p_{\mathrm{SAT}}(\cdot|x)$, if $y^+, y^-$ are conditionally i.i.d. from $\pi_{\hat{\theta}}(\cdot|x)$ and $\mathbb{E}\|d_{\hat{\theta}}\|^2 < \infty$, then for $h(y) := \ln \pi_{\hat{\theta}}(y \mid x) - \ln \pi_{\mathrm{ref}}(y \mid x)$,*

$$\left\| \mathbb{E}\big[\beta\, \sigma(-s)\, d_{\hat{\theta}}\big] \right\| \leq \frac{\beta}{4}\, \sqrt{\mathrm{Var}(s)}\, \sqrt{\mathbb{E}\|d_{\hat{\theta}}\|^2}, \qquad \mathrm{Var}(s) = 2\beta^2\, \mathrm{Var}_{Y \sim p_{\mathrm{SAT}}}\big(h(Y)\big). \qquad (10)$$

*Proof.* By (A5), for each $x$ we have $\mathbb{E}[d_{\hat{\theta}} \mid x] = 0$ since $y^+, y^-$ are i.i.d. under $\pi_{\hat{\theta}}(\cdot|x)$. Therefore

$$\mathbb{E}\big[\beta\, \sigma(-s)\, d_{\hat{\theta}}\big] = \beta\, \mathbb{E}\Big[\big(\sigma(-s) - \mathbb{E}\sigma(-s)\big)\, d_{\hat{\theta}}\Big].$$

Fix any unit vector $u$. Scalar Cauchy–Schwarz yields

$$u^\top \mathbb{E}\big[\beta\, \sigma(-s)\, d_{\hat{\theta}}\big] = \beta\, \mathbb{E}\Big[\big(\sigma(-s) - \mathbb{E}\sigma(-s)\big)\, u^\top d_{\hat{\theta}}\Big] \leq \beta\, \sqrt{\mathrm{Var}(\sigma(-s))}\, \sqrt{\mathbb{E}\big[(u^\top d_{\hat{\theta}})^2\big]}.$$

Taking the supremum over all unit $u$,

$$\left\| \mathbb{E}\big[\beta\, \sigma(-s)\, d_{\hat{\theta}}\big] \right\| \leq \beta\, \sqrt{\mathrm{Var}(\sigma(-s))}\, \sqrt{\mathbb{E}\|d_{\hat{\theta}}\|^2}.$$

Since $\sigma$ is $1/4$-Lipschitz, $\mathrm{Var}(\sigma(-s)) \leq \frac{1}{16}\, \mathrm{Var}(s)$. With $s = \beta[h(y^+) - h(y^-)]$ and $y^+, y^-$ i.i.d.,

$$\mathrm{Var}(s) = \beta^2\, \mathrm{Var}\big(h(y^+) - h(y^-)\big) = 2\,\beta^2\, \mathrm{Var}_{Y \sim p_{\mathrm{SAT}}}\big(h(Y)\big),$$

which gives the stated inequality. $\qquad\qquad\square$

# C ADDITIONAL RESULTS

## C.1 WILDBENCH LEADERBOARD FULL TABLE

Table 4: Complete WildBench leaderboard showing all evaluated models with comprehensive task-specific scores across multiple evaluation dimensions. Our models are highlighted in gray.

| Model | Elo | Task | Creative | Reasoning | Math | Info Seek | Coding | Length |
|---|---|---|---|---|---|---|---|---|
| GPT-4o (2024-05-13) | 1256.86 | 59.30 | 59.12 | 60.21 | 57.29 | 58.61 | 60.47 | 3723.52 |
| Claude-3.5-Sonnet (20240620) | 1238.93 | 54.70 | 55.61 | 55.64 | 50.16 | 55.54 | 56.51 | 2911.85 |
| GPT-4-Turbo (2024-04-09) | 1233.99 | 55.22 | 58.66 | 56.20 | 51.00 | 57.18 | 55.07 | 3093.17 |
| Gemini-1.5-Pro | 1228.55 | 52.95 | 55.12 | 53.73 | 48.59 | 52.23 | 55.22 | 3247.97 |
| DeepSeek-v2-Chat (0628) | 1221.66 | 53.99 | 56.43 | 54.83 | 51.43 | 52.72 | 55.00 | 3252.38 |
| GPT-4 (0125-preview) | 1221.30 | 52.28 | 57.57 | 53.45 | 45.79 | 54.36 | 52.92 | 3335.64 |
| Claude-3-Opus (20240229) | 1219.27 | 51.71 | 53.02 | 52.53 | 46.75 | 53.47 | 53.30 | 2685.98 |
| Qwen2.5-14B-UltraFeedback-DRIFT-iter2 | 1218.75 | 59.27 | 59.48 | 61.05 | 59.44 | 58.66 | 57.64 | 4275.27 |
| Mistral-Large-2 | 1217.99 | 55.57 | 58.86 | 57.22 | 52.67 | 57.38 | 53.84 | 3503.63 |
| Qwen2.5-14B-WildFeedback-DRIFT-iter2 | 1217.61 | 58.30 | 58.14 | 60.60 | 59.37 | 58.37 | 55.19 | 4492.77 |
| GPT-4o-mini (2024-07-18) | 1217.35 | 57.14 | 60.05 | 58.24 | 54.05 | 57.43 | 57.17 | 3648.13 |
| Qwen2.5-14B-WildFeedback-DRIFT-iter1-4k | 1215.73 | 58.37 | 58.55 | 60.39 | 58.89 | 58.86 | 55.57 | 4528.15 |
| Qwen2.5-14B-UltraFeedback-DRIFT-iter1 | 1215.67 | 58.52 | 58.65 | 60.30 | 58.02 | 57.57 | 57.64 | 4288.24 |
| Qwen2.5-14B-UltraFeedback-IterDPO-iter2 | 1215.12 | 54.78 | 55.35 | 56.76 | 54.00 | 55.00 | 53.08 | 3152.61 |
| Qwen2.5-14B-UltraFeedback-IterDPO-iter1 | 1214.14 | 54.21 | 55.13 | 56.27 | 52.02 | 55.74 | 52.64 | 3424.60 |
| Qwen2.5-14B-WildFeedback-DRIFT-iter2-4k | 1214.03 | 57.59 | 57.98 | 59.73 | 57.45 | 57.62 | 55.38 | 5333.00 |
| Qwen2.5-14B-WildFeedback-Seed | 1213.50 | 54.93 | 54.94 | 57.30 | 53.60 | 55.45 | 53.30 | 3878.89 |
| Qwen2.5-14B-Instruct | 1213.17 | 55.08 | 55.71 | 57.84 | 54.98 | 54.95 | 52.23 | 3682.95 |
| Qwen2.5-14B-WildFeedback-DRIFT-iter1 | 1212.83 | 57.27 | 58.09 | 59.31 | 55.78 | 57.22 | 56.13 | 4485.35 |
| Qwen2.5-14B-WildFeedback-IterDPO-iter2-4k | 1211.69 | 56.63 | 56.18 | 59.19 | 56.51 | 56.78 | 54.25 | 4491.05 |
| DeepSeek-v2-Coder (0628) | 1206.89 | 45.66 | 40.78 | 47.17 | 46.43 | 40.05 | 48.87 | 2580.18 |
| Gemini-1.5-Flash | 1206.77 | 48.85 | 51.66 | 50.79 | 45.32 | 48.67 | 48.73 | 3654.40 |
| Qwen2.5-14B-WildFeedback-IterDPO-iter1-4k | 1206.65 | 51.79 | 50.75 | 54.49 | 51.24 | 52.67 | 49.43 | 3925.74 |
| Qwen2.5-14B-WildFeedback-IterDPO-iter2 | 1206.63 | 51.38 | 50.23 | 54.17 | 51.03 | 52.48 | 48.68 | 4011.82 |
| Qwen2.5-14B-WildFeedback-IterDPO-iter1 | 1205.48 | 52.34 | 52.87 | 54.97 | 51.71 | 54.06 | 48.96 | 4054.28 |
| DeepSeek-v2-Chat | 1205.02 | 48.21 | 53.59 | 50.63 | 44.52 | 51.81 | 44.43 | 2896.97 |
| Qwen2.5-14B-WildFeedback-SPIN-iter1 | 1200.63 | 47.16 | 49.90 | 49.75 | 47.25 | 47.67 | 43.11 | 2707.18 |
| Qwen2.5-7B-WildFeedback-DRIFT-iter2 | 1199.09 | 51.69 | 52.45 | 53.21 | 50.63 | 52.38 | 50.28 | 4707.48 |
| Qwen2.5-7B-UltraFeedback-DRIFT-iter2 | 1197.94 | 50.32 | 50.39 | 52.50 | 48.41 | 52.08 | 48.58 | 5121.20 |
| Qwen2.5-7B-WildFeedback-DRIFT-4k-iter1 | 1197.13 | 51.06 | 53.23 | 53.44 | 48.32 | 52.28 | 49.29 | 4517.02 |
| Qwen2.5-7B-UltraFeedback-DRIFT-iter1 | 1197.04 | 50.91 | 50.08 | 53.77 | 48.92 | 52.72 | 48.87 | 4856.83 |
| Qwen2.5-7B-WildFeedback-DRIFT-4k-iter2 | 1195.33 | 51.06 | 51.42 | 53.45 | 49.16 | 52.13 | 49.38 | 4686.39 |
| Qwen2.5-7B-WildFeedback-DRIFT-iter1 | 1194.81 | 50.61 | 52.09 | 53.03 | 48.56 | 52.13 | 48.34 | 4652.38 |
| Qwen2.5-7B-Instruct | 1194.67 | 48.66 | 50.08 | 51.80 | 47.09 | 50.69 | 45.00 | 4275.08 |
| Qwen2.5-7B-UltraFeedback-IterDPO-iter1 | 1194.45 | 48.77 | 49.35 | 51.39 | 46.03 | 50.89 | 46.82 | 3888.79 |
| Qwen2.5-7B-WildFeedback-DRIFT-iter2-RPO | 1194.24 | 50.42 | 51.89 | 53.31 | 48.88 | 51.49 | 47.52 | 4711.01 |
| Qwen2.5-7B-WildFeedback-Seed | 1193.66 | 49.11 | 49.35 | 51.57 | 47.54 | 50.15 | 47.17 | 4624.30 |
| Nemotron-4-340B-Instruct | 1193.60 | 47.67 | 53.32 | 49.13 | 40.80 | 53.00 | 46.26 | 2754.01 |
| Qwen2.5-7B-WildFeedback-SPIN-iter2 | 1192.56 | 44.04 | 45.54 | 45.95 | 40.64 | 45.89 | 43.13 | 2820.38 |
| Qwen2.5-7B-WildFeedback-IterDPO-iter2-4k | 1192.46 | 48.94 | 50.39 | 51.35 | 46.37 | 50.99 | 46.79 | 4731.32 |
| Claude-3-Sonnet (20240229) | 1192.03 | 45.48 | 46.30 | 47.43 | 40.64 | 47.13 | 46.10 | 2670.24 |
| Qwen2.5-7B-UltraFeedback-IterDPO-iter2 | 1192.01 | 48.49 | 50.18 | 50.73 | 44.68 | 50.22 | 47.58 | 3708.61 |
| Qwen2-72B-Instruct | 1189.43 | 44.50 | 49.92 | 46.85 | 40.95 | 49.50 | 39.81 | 2856.45 |
| Qwen2.5-7B-WildFeedback-IterDPO-iter1-4k | 1189.43 | 47.07 | 48.63 | 49.13 | 44.50 | 49.26 | 45.12 | 4602.08 |
| Mistral-Nemo-Instruct (2407) | 1187.35 | 44.38 | 54.57 | 47.41 | 35.63 | 51.93 | 39.72 | 3318.21 |
| Qwen2.5-7B-WildFeedback-IterDPO-iter1 | 1185.11 | 46.31 | 46.77 | 48.61 | 44.46 | 48.02 | 44.27 | 4662.34 |
| Qwen2.5-7B-WildFeedback-IterDPO-iter2 | 1182.33 | 46.17 | 45.74 | 49.22 | 45.60 | 46.24 | 43.70 | 4520.58 |
| Qwen2.5-7B-WildFeedback-SPIN-iter1 | 1180.75 | 42.86 | 43.26 | 45.45 | 41.59 | 46.29 | 39.06 | 3611.19 |
| Qwen2.5-14B-UltraFeedback-SPIN-iter1 | 1178.88 | 36.99 | 38.86 | 40.09 | 35.16 | 39.01 | 33.40 | 2642.64 |
| Claude-3-Haiku (20240307) | 1175.97 | 38.89 | 42.95 | 41.29 | 31.43 | 45.35 | 36.98 | 2601.03 |
| Qwen2.5-7B-WildFeedback-SPIN-iter2 | 1173.45 | 37.86 | 37.36 | 40.72 | 37.06 | 42.13 | 33.27 | 2839.09 |
| Mistral-Large (2402) | 1172.18 | 38.89 | 49.66 | 41.80 | 30.88 | 46.14 | 33.74 | 2514.98 |
| Qwen2.5-7B-UltraFeedback-SPIN-iter1 | 1163.16 | 35.10 | 34.78 | 37.61 | 30.87 | 39.90 | 33.21 | 3475.16 |
| Qwen1.5-72B-Chat-Greedy | 1160.02 | 39.93 | 50.36 | 43.45 | 29.80 | 48.22 | 35.36 | 2392.36 |
| Qwen2.5-14B-UltraFeedback-SPIN-iter2 | 1155.07 | 28.01 | 29.77 | 31.69 | 25.08 | 32.72 | 23.13 | 2504.57 |
| Mixtral-8x7B-Instruct-v0.1 | 1145.51 | 31.47 | 42.75 | 34.59 | 22.14 | 41.94 | 25.02 | 2653.58 |
| Qwen2.5-7B-UltraFeedback-SPIN-iter2 | 1139.66 | 25.93 | 25.79 | 28.88 | 19.60 | 32.03 | 24.43 | 3293.30 |
| GPT-3.5-Turbo (0125) | 1135.69 | 30.02 | 37.42 | 33.39 | 21.59 | 36.49 | 26.54 | 1844.14 |

## C.2 UNGUIDED ABLATION STUDY

Table 5: Unguided ablation study results on Qwen2.5-7B (full *WildFeedback* setting). Unguided variants remove the reference DSAT response and explicit improvement instruction, matching SPIN's setting.

| Method | WildBench Score | AlpacaEval2 WinRate |
|---|---|---|
| *SPIN* | | |
| iter1 | 42.86 | 26.21 |
| iter2 | 37.86 | 20.56 |
| *IterDPO* | | |
| iter1 (Unguided) | 49.44 | 40.30 |
| iter2 (Unguided) | 49.66 | 41.24 |
| iter1 (Full) | 46.31 | 40.36 |
| iter2 (Full) | 46.17 | 35.76 |
| *DRIFT (Ours)* | | |
| iter1 (Unguided) | 50.77 | **46.77** |
| iter2 (Unguided) | 51.04 | 45.47 |
| iter1 (Full) | 50.61 | 43.90 |
| iter2 (Full) | **51.69** | 46.64 |

## C.3 GENERALIZATION ACROSS MODEL FAMILIES

Table 6: Experiment results on Gemma-3-12B-it (full *WildFeedback* setting). All methods were trained using the same recipe and evaluated on WildBench. Bold indicates the highest score for each metric.

| Method | Task | Creative | Reasoning | Math | Info Seek | Coding |
|---|---|---|---|---|---|---|
| Base | 60.77 | 66.10 | 62.46 | 54.10 | **65.21** | 59.62 |
| *SPIN* | | | | | | |
| iter1 | 59.06 | 65.79 | 61.14 | 50.40 | 63.96 | 58.30 |
| iter2 | 51.86 | 58.76 | 53.81 | 42.06 | 59.60 | 50.33 |
| *IterDPO* | | | | | | |
| iter1 | 60.10 | 62.74 | 60.48 | 54.92 | 61.29 | **62.10** |
| iter2 | 48.98 | 52.40 | 49.27 | 43.89 | 47.61 | 52.23 |
| *DRIFT (Ours)* | | | | | | |
| iter1 | **61.73** | **66.37** | **63.02** | **56.19** | 64.71 | 61.23 |
| iter2 | 60.88 | 66.10 | 62.07 | 54.76 | 62.82 | 61.33 |

## C.4 SAFETY AND ETHICAL CONSIDERATIONS

Table 7: Safety and ethical norms evaluation results. AdvBench measures adversarial jailbreak attack success rate (%). ToxiGen measures toxicity scores (1–5 scale). Lower values indicate better safety.

| Method | Attack Success Rate (%) | Toxic Score (1-5) |
|---|---|---|
| Base | 0.01 | 1.56 |
| *SPIN* | | |
| iter1 | 0.01 | 1.65 |
| iter2 | 0.00 | 1.70 |
| *IterDPO* | | |
| iter1 | 0.02 | 1.50 |
| iter2 | 0.00 | 1.66 |
| *DRIFT (Ours)* | | |
| iter1 | 0.02 | 1.54 |
| iter2 | 0.01 | 1.58 |

# D   IMPLEMENTATION DETAILS

## D.1   WARM START TRAINING DETAILS

We curate a DSAT→SAT seed set (491 pairs) from WildFeedback, where a dissatisfied user turn (DSAT) is followed by a revised model response that satisfies the user (SAT). Each pair provides a natural preference: the DSAT response fails to meet expectations, while the subsequent SAT response is preferred.

For our warm start phase, we initialize training using pre-trained instruction-tuned models as the base models. The warm start training utilizes seed preference data to establish initial alignment before iterative refinement. We did DPO training with carefully tuned hyperparameters to ensure stable convergence. All experiments were conducted on 8 H100 GPUs with the same hardware configuration maintained across all training phases. The detailed hyperparameters for warm start training are presented in Table 8.

Table 8: Warm Start Training Hyperparameters

| Learning rate | Batch size | $\beta$ | Optimizer | LR scheduler | Seq length | Epochs | Precision |
|---|---|---|---|---|---|---|---|
| 5.0e-7 | 4 | 0.1 | RMSprop | Linear | 2048 | 3 | bfloat16 |

## D.2   ITERATIVE TRAINING DETAILS

After the warm start phase, we conducted iterative training to progressively refine model alignment using dynamically generated preference data. Data generation details are in Sec. 4.1. Each iteration builds upon the previous model checkpoint, incorporating newly created preference data. The iterative training process maintains consistent hyperparameters across iterations, with only the training data and base model checkpoint changing between iterations. We trained each iteration for a single epoch to prevent overfitting on the iteratively generated data. Table 9 details the hyperparameters used for iterative training phases.

Table 9: Iterative Training Hyperparameters

| Learning rate | Batch size | $\beta$ | Optimizer | LR scheduler | Seq length | Epochs | Precision |
|---|---|---|---|---|---|---|---|
| 5.0e-7 | 4 | 0.1 | RMSprop | Linear | 2048 | 1 | bfloat16 |

## D.3   TRAINING DYNAMICS

For better training illustration, we report the Qwen2.5-14B-Instruct DRIFT iter1 & iter2 training dynamics in Figure 5 which shows dpo training loss, chosen reward, and rejected reward. The loss curves exhibit stable convergence across both iterations. The reward signals show the expected separation pattern: chosen rewards consistently increase while rejected rewards decrease. This trend is observed in both iterations, confirming the effectiveness of the training.

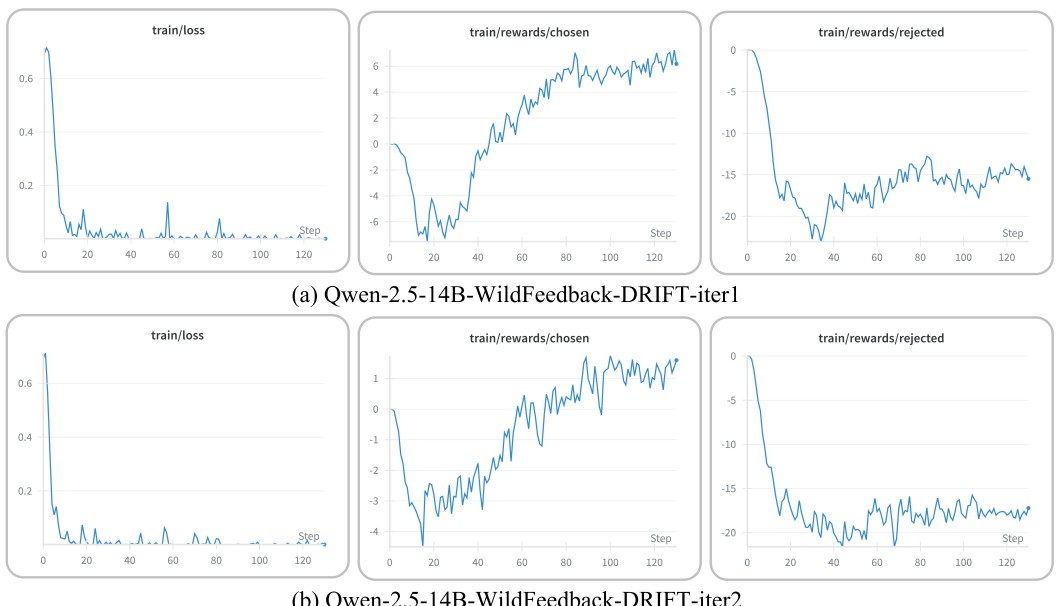

(a) Qwen-2.5-14B-WildFeedback-DRIFT-iter1

(b) Qwen-2.5-14B-WildFeedback-DRIFT-iter2

Figure 5: The top row shows DRIFT training dynamics for iteration 1 on Qwen2.5-14B-Instruct. The bottom row shows the training dynamics for iteration 2.

## E   MODEL RESPONSES EXAMPLE

**Prompt:** Using at least two different scientific sources, generate a research paper that analyzes and compares the form and function of three different types of carnivorous plants, including but not limited to Venus flytraps, pitcher plants, and sundews. The paper should include detailed descriptions of the physical characteristics of each plant, as well as an analysis of their unique adaptations for capturing and digesting prey. In addition, the paper should explore the evolutionary history and ecological significance of carnivorous plants, and discuss current research on their use in medicine and biotechnology. The paper should be written in APA format and include at least 10 peer-reviewed references.

**SPIN Representative Response:**
Title: Carnivorous Plants: Form, Function, Evolutionary History, and Applications

Introduction:
Carnivorous plants are fascinating organisms that have evolved unique adaptations to capture and digest prey. These plants are found in nutrient-poor environments where soil lacks sufficient nitrogen and phosphorus. The three most common types of carnivorous plants are Venus flytraps (Dionaea muscipula), pitcher plants (Nepenthes spp.), and sundews (Drosera spp.). This research paper will analyze and compare the form and function of these three types of carnivorous plants, as well as their evolutionary history and ecological significance.

Physical Characteristics and Adaptations:
Venus flytraps are known for their trap-like leaves that can close rapidly when triggered by insects. Each leaf has six trigger hairs that, when touched twice within 20 seconds, cause the trap to close. Once closed, the trap secretes digestive enzymes to break down the insect. Venus flytraps are native to North Carolina and South Carolina and require specific environmental conditions, such as high humidity and acidic soil, to thrive.

Pitcher plants are named for their modified leaves that resemble pitchers or cups. These leaves are covered in nectar and have a slippery surface that makes it difficult for insects to escape once they enter. Once inside, the insect is digested by enzymes secreted by the plant. Pitcher plants are found throughout the world, from tropical rainforests to temperate regions, and come in various shapes and sizes. Nepenthes rafflesiana, also known as the Rafflesia pitcher plant, is one of the largest pitcher plants, with pitchers up to 30 centimeters tall.

Sundews are named for their sticky glandular hairs that resemble dewdrops. When an insect lands on the plant, the sticky hairs trap it, and the plant secretes digestive enzymes to break down the insect. Sundews are found in many parts of the world, including North America, Europe, and Asia. Drosera capensis, also known as the Cape sundew, is a popular species used in horticulture due to its attractive appearance and ease of care.

Evolutionary History and Ecological Significance:
The evolutionary history of carnivorous plants is still not fully understood, but it is believed that they evolved independently multiple times across different plant lineages. Carnivorous plants likely evolved in nutrient-poor environments as a way to supplement their nutrition through the capture and digestion of insects.
...

**GPT-5 Score: 5/10**
**Explanation:** Covers the basic features of the plants but lacks depth; references are not standardized (limited to general descriptions rather than APA format); missing an abstract and overall not well structured as a research paper.

Figure 6: SPIN model response example.

**Prompt:** Using at least two different scientific sources, generate a research paper that analyzes and compares the form and function of three different types of carnivorous plants, including but not limited to Venus flytraps, pitcher plants, and sundews. The paper should include detailed descriptions of the physical characteristics of each plant, as well as an analysis of their unique adaptations for capturing and digesting prey. In addition, the paper should explore the evolutionary history and ecological significance of carnivorous plants, and discuss current research on their use in medicine and biotechnology. The paper should be written in APA format and include at least 10 peer-reviewed references.

**IterDPO Representative Response:**
Title: Carnivorous Plants: Form, Function, and Evolutionary Significance

Abstract:
Carnivorous plants are fascinating organisms that have evolved unique mechanisms to capture and digest prey. This paper compares the form and function of three different types of carnivorous plants: Venus flytraps (Dionaea muscipula), pitcher plants (Nepenthes spp.), and sundews (Drosera spp.). We also discuss the evolutionary history and ecological significance of these plants, as well as their potential applications in medicine and biotechnology.

Introduction:
Carnivorous plants are found in nutrient-poor environments where other sources of nitrogen and phosphorus are scarce. They have evolved specialized trapping mechanisms to obtain these nutrients from insects and small animals. Venus flytraps, pitcher plants, and sundews are three examples of carnivorous plants with distinct trapping strategies.

Venus Flytraps:
Venus flytraps are native to North Carolina and South Carolina in the United States. Their leaves contain trigger hairs that, when touched twice, cause the leaf to snap shut, trapping the insect inside (Gurevitch et al., 2017). Once trapped, the insect is digested by enzymes secreted by the plant. Venus flytraps have evolved this mechanism to supplement their nutrient intake in nutrient-poor soil (Hunt & Raker, 2018).

Pitcher Plants:
Pitcher plants are found in tropical and subtropical regions worldwide. They have evolved a modified leaf that forms a deep, liquid-filled pitfall trap (Kress & Wurdack, 2006). Insects are attracted to the nectar produced by the pitcher and fall into the liquid, where they are digested by enzymes and bacteria. Pitcher plants have evolved this mechanism to obtain nutrients from a wide range of prey, including ants, flies, and spiders (Rice, 2014).

Sundews:
Sundews are found in temperate and tropical regions worldwide. They have evolved sticky glands on their leaves that trap insects (Orozco-terWengel & Brodersen, 2015). Once trapped, the insect is digested by enzymes secreted by the plant. Sundews have evolved this mechanism to obtain nutrients from a wide range of prey, including flies, mosquitoes, and ants (Rice, 2014).
...

**GPT-5 Score: 6/10**
**Explanation:** Presents a clearer structure with some academic tone, but the discussion is still surface-level and misses key elements like applications and conclusion.

Figure 7: IterDPO model response example.

**Prompt:** Using at least two different scientific sources, generate a research paper that analyzes and compares the form and function of three different types of carnivorous plants, including but not limited to Venus flytraps, pitcher plants, and sundews. The paper should include detailed descriptions of the physical characteristics of each plant, as well as an analysis of their unique adaptations for capturing and digesting prey. In addition, the paper should explore the evolutionary history and ecological significance of carnivorous plants, and discuss current research on their use in medicine and biotechnology. The paper should be written in APA format and include at least 10 peer-reviewed references.

**DRIFT Representative Response:**
**Title: Comparative Analysis of Carnivorous Plant Adaptations: Venus Flytraps, Pitcher Plants, and Sundews**

**Abstract**
Carnivorous plants have evolved remarkable adaptations to survive in nutrient-poor environments by capturing and digesting insects. This paper compares the form and function of three prominent carnivorous plants: *Dionaea muscipula* (Venus flytrap), *Nepenthes spp.* (pitcher plants), and *Drosera spp.* (sundews). We analyze their physical characteristics, trapping mechanisms, and digestive processes. Additionally, we discuss their evolutionary history, ecological roles, and potential applications in medicine and biotechnology. Our review synthesizes data from multiple studies to provide a comprehensive understanding of these fascinating organisms.

**Introduction**
Carnivorous plants are specialized angiosperms that have developed unique strategies to obtain nutrients from animal prey (Lloyd, 1983). They thrive in nutrient-poor soils, where traditional photosynthetic pathways alone cannot sustain growth. Three notable examples of carnivorous plants are the Venus flytrap (*Dionaea muscipula*), pitcher plants (*Nepenthes spp.*), and sundews (*Drosera spp.*). Each species has distinct morphological features and physiological mechanisms adapted for prey capture and digestion.

**Venus Flytrap (*Dionaea muscipula*)**
The Venus flytrap is characterized by its hinged leaves with sensitive trigger hairs. When an insect touches these hairs, the leaf rapidly closes within seconds (Braun & Speck, 2006). The internal structure includes a complex system of mechanical sensors and hydraulic pressures that facilitate rapid closure (Lischka et al., 2014).
**Pitcher Plants (*Nepenthes spp.*)**
Pitcher plants possess modified leaves forming a deep, fluid-filled trap. Insects are attracted to the nectar produced at the pitcher's entrance and fall into the digestive fluid below (Givnish, 2000). The pitchers contain enzymes and acids that break down the prey (Bramwell & Chaloner, 1975).
**Sundews (*Drosera spp.*)**
Sundews are covered in sticky glandular hairs that ensnare insects. Once an insect becomes stuck, the plant secretes digestive enzymes to dissolve the prey (O'Leary et al., 2010).
...

**GPT-5 Score: 8/10**
**Explanation:** The format is closer to an academic paper (title, abstract, introduction, subsections) and using markdown format; covers morphology, function, evolution, ecology, and applications; relatively comprehensive and consistent with a review style.

Figure 8: DRIFT model response example.

