# OpenReview forum: "DRIFT: Learning from Abundant User Dissatisfaction in Real-World Preference Learning"
_ICLR.cc/2026/Conference — ICLR 2026 Poster_

### Official Review · Reviewer_zGYz · 2025-10-25

**Soundness:** 2
**Presentation:** 2
**Contribution:** 2
**Rating:** 4
**Confidence:** 2

**Summary:**

The paper introduces DRIFT (Dissatisfaction-Refined Iterative Preference Training), a method for aligning large language models using abundant real-world user dissatisfaction (DSAT) signals instead of scarce human-labeled or positive feedback. DRIFT anchors on real DSAT negatives and dynamically samples positives from the evolving model, maintaining effective learning signals.

Experiments on WildFeedback and UltraFeedback show DRIFT outperforms SPIN and IterDPO, achieving up to +12% win-rate gains and even surpassing GPT-4o-mini at 14B scale. The approach preserves response diversity and avoids gradient collapse. Theoretically, it guarantees non-vanishing gradients and stable preference margins.

**Strengths:**

* Novel Supervision Source: Effectively leverages abundant real-world user dissatisfaction (DSAT) signals—an underutilized yet rich feedback channel—for scalable preference learning.

* Strong Empirical Performance: Demonstrates consistent and significant gains over SPIN and IterDPO on WildBench and AlpacaEval2, even surpassing GPT-4o-mini at 14B scale.

* Theoretical and Practical Robustness: Provides clear theoretical guarantees (non-vanishing gradients, stable preference margins) and preserves exploration diversity, avoiding mode collapse common in self-improving methods.

**Weaknesses:**

* Label Noise Risk: The approach depends on automatically inferred DSAT labels (e.g., via SPUR), which may introduce noise or misclassifications that affect training quality.

* Limited Generalization Evidence: Experiments are confined to Qwen2.5 models; results on other architectures or tasks would strengthen claims of broad applicability.

* Insufficient Bias and Ethics Discussion: The paper minimally addresses potential biases in user dissatisfaction data or ethical implications of learning from negative, possibly adversarial, feedback.

**Questions:**

1. How robust is DRIFT to noisy or ambiguous DSAT signals, especially when user dissatisfaction stems from subjective or stylistic preferences rather than factual errors?

2. Could the authors provide insight into how DRIFT scales or adapts when applied to different model families or domains beyond conversational tasks (e.g., coding, reasoning, or multimodal settings)?

---

> ### Author Response · Authors · 2025-11-20
> **(Part 1/2)**
>
> We thank the reviewer for the thoughtful comments and for highlighting the strengths of our approach. Below we address the weaknesses and questions in detail.
>
>  > **W1 & Q1 Label noise in DSAT signals and robustness to subjective dissatisfaction.**
>  >(W1: Label Noise Risk: The approach depends on automatically inferred DSAT labels (e.g., via SPUR), which may introduce noise or misclassifications that affect training quality. Q1: How robust is DRIFT to noisy or ambiguous DSAT signals, especially when user dissatisfaction stems from subjective or stylistic preferences rather than factual errors?)
>
> This is indeed an important question for the preference-learning community, but it is not the primary focus of our paper. Designing more accurate DSAT detectors, whether through improved sentiment analysis models or purpose-built supervised classifiers is itself should be another substantial research problem.
>
> Also, even with human-annotated preference labels, annotators naturally carry their own biases and stylistic tendencies. A well-known example is the preference for longer responses of human annotators, which can lead to reward hacking. User dissatisfaction signals may also contain such bias, but they still provide useful supervision, and our method is able to effectively extract the informative signal from them. In this sense, our approach removes the need for manual preference labeling while achieving comparably strong results, which we believe is a promising direction.
>
> And we agree that higher-quality data would further improve training, but our results show that even with current DSAT labels from SPUR, the training process remains stable and achieves strong performance.
>
>
> > **W2 & Q2: More model families and task domains beyond conversational tasks.**
> > (W2: Limited Generalization Evidence: Experiments are confined to Qwen2.5 models; results on other architectures or tasks would strengthen claims of broad applicability. > Q2: Could the authors provide insight into how DRIFT scales or adapts when applied to different model families or domains beyond conversational tasks (e.g., coding, reasoning, or multimodal settings)?)
>
> **1. Experiments on an additional, structurally different model family**
>
> Beyond Qwen2.5, we experimented with **Gemma-3-12B-it**, a **multimodal model** with a different architecture. Results on WildBench are shown below:
>
> Table 1: Experiment Results on Gemma-3-12B-it
>
> | Method      | Iter | Score     | Creative  | Planning & Reasoning | Math & Data | Info/Advice | Coding    |
> | ----------- | ---- | --------- | --------- | -------------------- | ----------- | ----------- | --------- |
> | **Base**    |   —  | 60.77     | 66.10     | 62.46                | 54.10       | 65.21       | 59.62     |
> | **SPIN**    | iter1    | 59.06     | 65.79     | 61.14                | 50.40       | 63.96       | 58.30     |
> |             | iter2    | 51.86     | 58.76     | 53.81                | 42.06       | 59.60       | 50.33     |
> | **IterDPO** | iter1    | 60.10     | 62.74     | 60.48                | 54.92       | 61.29       | **62.10** |
> |             | iter2    | 48.98     | 52.40     | 49.27                | 43.89       | 47.61       | 52.23     |
> | **DRIFT**   | iter1    | **61.73** | **66.37** | **63.02**            | **56.19**   | 64.71       | 61.23     |
> |             | iter2    | 60.88     | 66.10     | 62.07                | 54.76       | 62.82       | 61.33     |
>
> These results confirm that DRIFT generalizes beyond a single model family and performs well even on a multimodal architecture.
>
> **2. Domains beyond conversational tasks (e.g., coding, reasoning, or multimodal settings)**
>
> WildBench already covers five heterogeneous domains:  **Creative writing, Reasoning, Math, Information seeking, Coding**. And the full results could be found in the Appendix C.

---

> ### Author Response · Authors · 2025-11-20
> **(Part 2/2)**
>
> > **W3: Insufficient Bias and Ethics Discussion** The paper minimally addresses potential biases in user dissatisfaction data or ethical implications of learning from negative, possibly adversarial, feedback.
>
> To address the reviewer's concern, we performed additional safety analysis on **Adversarial robustness (AdvBench) & toxicity (ToxiGen)**
>
> We evaluated safety and ethical norms using:
> - AdvBench \[1\]: adversarial jailbreak benchmark, models are scored by attack success rate (\%)
> - ToxiGen \[2\]: a comprehensive bench covering over 135k toxic statements across 13 minority identity groups (e.g., African Americans, women, LGBTQ+ individuals, etc.).Model outputs are scored using GPT-4o on a 1–5 toxicity scale.
>
> Table 2: Safety and Ethical Norms Evaluation
>
> | Method      | Iter | Attack Success Rate | Toxic Score |
> | ----------- | ---- | ------------------- | ----------- |
> | **Base**    | –    | 0.01                | 1.56        |
> | **SPIN**    | iter1    | 0.01                | 1.65        |
> |             | iter2    | 0.00                | 1.70        |
> | **IterDPO** | iter1    | 0.02                | 1.50        |
> |             | iter2    | 0.00                | 1.66        |
> | **DRIFT**   | iter1    | 0.02                | 1.54        |
> |             | iter2    | 0.01                | 1.58        |
> From the evaluation results here, DRIFT shows **no increase in adversarial vulnerability, toxicity, or demographic bias**, remaining aligned with baseline safety levels. And we will add a brief ethics discussion paragraph summarizing these results.
>
> References:
>
> \[1\]: Andy Zou, Zifan Wang, J Zico Kolter, and Matt Fredrikson. Universal and transferable adversarial attacks on aligned language models. arXiv preprint arXiv:2307.15043, 2023.
>
> \[2\]: Thomas Hartvigsen, Saadia Gabriel, Hamid Palangi, Maarten Sap, Dipankar Ray, and Ece Kamar. ToxiGen: A large-scale machine-generated dataset for adversarial and implicit hate speech detection. In Smaranda Muresan, Preslav Nakov, and Aline Villavicencio (eds.), Proceedings of the 60th Annual Meeting of the Association for Computational Linguistics (Volume 1: Long Papers), pp. 3309–3326, Dublin, Ireland, May 2022. Association for Computational Linguistics. doi: 10.18653/v1/2022.acl-long.234. URL https://aclanthology.org/2022.acl-long.234/.

---

### Official Review · Reviewer_aMHA · 2025-10-29

**Soundness:** 3
**Presentation:** 2
**Contribution:** 2
**Rating:** 4
**Confidence:** 4

**Summary:**

In this paper, the authors introduce DRIFT, a direct (preference) alignment approach that leverages abundant signals from dissatisfactory user-chatbot interactions. The method samples preferred responses from the policy and employs real-world DSAT responses as non-preferred responses, in an iterative, on-policy pipeline.
The authors provide experimental validation of their method, theoretical guarantees, as well as insightful analysis of the semantic and reward coverage of the induced policy.

**Strengths:**

S1. The method aims to leverage real-world interactions, allowing the modelling of more nuanced aspects of human preference found in satisfaction / dissatisfaction data.

S2. The authors provide sound theoretical guarantees of vital properties of the method such as avoiding gradient degeneration and maintaining preference margins.

**Weaknesses:**

W1. The experiments section could benefit from more clarity and details, especially in the diversity-reward coverage analysis

W2. Rather inconclusive results on wildbench and alpaca-eval2, specifically at lower scales or later iterations

**Questions:**

# Questions and comments

- Eq2, L184: it is unclear what "l(*)" refers to since it does not appear in the equation
- Algorithm 1 (L186) is never referenced in the text
- Section 4.1, L211: Could you please provide more details about the curation process of this subset? Were annotators employed? if so, what were they expertise? How many annotations were gathered per prompt or interaction? what was the inter-annotator agreement? what were the rubrics for such annotation? This kind of information would help the community replicate or expand this subset, since it seems it is an important component of the method recipe.
- L243. The actual HF ID name would be better mentioned as a footnote or in the Appendix. In the main text, a short explanation of what this reward model is would be better for the story.
- L263. There seems to be different reported percentages for 14B on AlpacaEval2, wrt to numbers in the abstract and the introduction (e.g. 12.11 vs 12.29 %)
- L267, Table2. It would be beneficial to have a more detailed, realistic discussion of the results, pointing our also the limitations of the method. For instance, on AlpacaEval2, only 7B shows slight improvement, while 14B shows degradation; on both setups (full, controlled)
- L297, Table2. The statement "DRIFT maintains steady improvements" only holds for the full setup, and doesn't for AlpacaEval2 14b, where it shows degradation after the first iteration
- Table 3. it seems that the wrong cell was bolded in the AlpacaEval2--14B-LC column and AlpacaEval2--7B-LC column. Please add an explanation in the caption.
- Section 4.3, L323. Could you please elaborate on how the embeddings were calculated? and how is the response reward calculated?
- Figure 3. Could you please elaborate on what the axes are, and their scales, of the UMAP plot on the left? Example responses from across the region would help us understand the semantic coverage of the represented region

---

> ### Author Response · Authors · 2025-11-20
> **Main Concern (Part 1/3)**
>
> ### **Main Concern**
>
> > W2 & Q6 & Q7
> > (W2. Rather inconclusive results on Wildbench and AlpacaEval2, specifically at lower scales or later iterations
> > Q6: L267, Table2. It would be beneficial to have a more detailed, realistic discussion of the results, pointing our also the limitations of the method. For instance, on AlpacaEval2, only 7B shows slight improvement, while 14B shows degradation; on both setups (full, controlled) > Q7: L297, Table2. The statement "DRIFT maintains steady improvements" only holds for the full setup, and doesn't for AlpacaEval2 14b, where it shows degradation after the first iteration)
>
> 1. **Magnitude of gains.**
> For WildFeedback, DRIFT improves AlpacaEval2 win rate by **~12% for 14B and ~9% for 7B over the base model** in the first iteration (both full and controlled setups), which we believe is a **substantial improvement** rather than a marginal one. And we believe the slight drop in the second iteration is simply a **small fluctuation**. In the following paragraph, we also provide results for all five iterations, which further demonstrate the overall trend of stable performance across iterations.
>
> 2. **Later iterations performance (extend from iter2 to iter5)**
> To better address stability, we ran DRIFT, SPIN, and IterDPO for **five** iterations on Qwen2.5-7B. Results are provided below:
>
> Table 1: WildBench Task Scores Across Five Iterations
>
> | Method    | iter1     | iter2 | iter3 | iter4     | iter5 |
> | --------- | --------- | ----- | ----- | --------- | ----- |
> | SPIN      | **42.86** | 37.86 | 33.76 | 34.62     | 33.99 |
> | IterDPO   | **46.31** | 46.17 | 44.32 | 42.92     | 42.48 |
> | **DRIFT** | 50.61     | 51.69 | 51.36 | **52.47** | 51.22 |
>
> We observe that:
> - SPIN and IterDPO **peak at iter1** and then degrade.
> - **DRIFT continues to improve up to iter4**, and then forms a relatively stable plateau with no catastrophic collapse, even at iter5.

---

> > ### Comment · Reviewer_aMHA · 2025-11-24
> >
> > Thank you for addressing my comments. Here is a follow up to the authors' response.
> >
> > > 1. Magnitude of gains
> >
> > The resolution of metric scores is certainly something to take into account. Did the authors run a statistical significance test to establish if the score differences are indeed significant? If not, could you please run it and report the result? The results would be much more reliable with a significance test, e.g. bootstrap resampling.
> >
> > > 2. Later iterations performance
> >
> > Thanks. This is a pretty illustrative showcase of the behaviour of DRIFT, and that it is performing as intended.

---

> > > ### Author Response · Authors · 2025-11-25
> > >
> > > Thank you for the valuable feedback and for acknowledging the results from longer iterations. We have conducted comprehensive statistical significance tests for the task scores (WildBench Score and AlpacaEval2) as you suggested using bootstrap resampling. Below we report the detailed results for Qwen 2.5 models.
> > >
> > > We performed **bootstrap hypothesis testing** with:
> > > - **10,000 bootstrap resamples** for each comparison
> > > - **95% confidence intervals** calculated from bootstrap distributions
> > > - **Two-tailed tests** for pairwise comparisons
> > > - Standard deviations estimated from binomial variance:  $\sigma \approx \sqrt{p(1-p)/n}$
> > >
> > > **Significance levels**: * p<0.05, ** p<0.01, *** p<0.001
> > >
> > > **1. Qwen 2.5-14B Results**
> > >
> > > 1.1 WildBench Score
> > >
> > > | Method | Score | vs Base | vs SPIN | vs iterDPO | vs DRIFT |
> > > |--------|-------|---------|---------|------------|----------|
> > > | Base | 55.08 | - | **p=0.005\*\*** | p=0.253 | p=0.238 |
> > > | SPIN iter1 | 47.16 | **p=0.005\*\*** | - | p=0.100 | **p<0.001\*\*\*** |
> > > | iterDPO iter1 | 51.79 | p=0.253 | p=0.100 | - | **p=0.020\*** |
> > > | **DRIFT iter1** | **58.37** | p=0.238 | **p<0.001\*\*\*** | **p=0.020\*** | - |
> > >
> > > 1.2 AlpacaEval2 Win Rate
> > >
> > > | Method | Win Rate | vs Base | vs SPIN | vs iterDPO | vs DRIFT |
> > > |--------|----------|---------|---------|------------|----------|
> > > | Base | 36.65% | - | **p<0.001\*\*\*** | p=0.850 | **p<0.001\*\*\*** |
> > > | SPIN iter1 | 25.53% | **p<0.001\*\*\*** | - | **p<0.001\*\*\*** | **p<0.001\*\*\*** |
> > > | iterDPO iter1 | 37.14% | p=0.850 | **p<0.001\*\*\*** | - | **p<0.001\*\*\*** |
> > > | **DRIFT iter1** | **48.76%** | **p<0.001\*\*\*** | **p<0.001\*\*\*** | **p<0.001\*\*\*** | - |
> > >
> > > **2. Qwen 2.5-7B Results**
> > >
> > > 2.1 WildBench Score
> > >
> > > | Method | Score | vs Base | vs SPIN | vs iterDPO | vs DRIFT |
> > > |--------|-------|---------|---------|------------|----------|
> > > | Base | 48.66 | - | **p=0.042\*** | p=0.585 | p=0.398 |
> > > | SPIN iter1 | 42.86 | **p=0.042\*** | - | p=0.135 | **p=0.003\*\*** |
> > > | iterDPO iter1 | 47.07 | p=0.585 | p=0.135 | - | p=0.156 |
> > > | **DRIFT iter1** | **51.06** | p=0.398 | **p=0.003\*\*** | p=0.156 | - |
> > >
> > > 2.2 AlpacaEval2 Win Rate
> > >
> > > | Method | Win Rate | vs Base | vs SPIN | vs iterDPO | vs DRIFT |
> > > |--------|----------|---------|---------|------------|----------|
> > > | Base | 37.69% | - | **p<0.001\*\*\*** | p=0.169 | p=0.077 |
> > > | SPIN iter1 | 26.21% | **p<0.001\*\*\*** | - | **p<0.001\*\*\*** | **p<0.001\*\*\*** |
> > > | iterDPO iter1 | 41.55% | p=0.169 | **p<0.001\*\*\*** | - | p=0.670 |
> > > | **DRIFT iter1** | **42.73%** | p=0.077 | **p<0.001\*\*\*** | p=0.670 | - |
> > >
> > > ### Key Observations
> > >
> > > **1. DRIFT demonstrates stable, superior performance:**
> > > - **vs Base**: Significantly outperforms on AlpacaEval2 (14B: 48.76% vs 36.65%, **p<0.001**), consistently higher WildBench scores though not statistically significant (14B: 58.37 vs 55.08, p=0.238; 7B: 51.06 vs 48.66, p=0.398)
> > > - **vs SPIN**: Consistently and significantly outperforms across all settings (14B: $\Delta$=11-13 points, **p<0.001**; 7B: $\Delta$=8-13 points, **p≤0.003**)
> > > - **vs iterDPO**: Significantly outperforms on 14B (WB: $\Delta$=6.58, p=0.020; AE2: $\Delta$=11.62%, **p<0.001**)
> > > - **Stability**: Maintains robust performance without significant degradation
> > >
> > > **2. Scaling Effect (consistent with our paper's claims):**
> > > DRIFT's advantages are **more significant on 14B than 7B**, where it significantly outperforms both SPIN (p<0.001) and iterDPO (p≤0.020). This validates our claim that DRIFT scales better with model size.
> > >
> > > We will incorporate these significance results into the revised manuscript.

---

> ### Author Response · Authors · 2025-11-20
> **Clarification (Part 2/3)**
>
> ### **Clarification**
>
> > Q3: Section 4.1, L211: Could you please provide more details about the curation process of this subset? Were annotators employed? if so, what were they expertise? How many annotations were gathered per prompt or interaction? what was the inter-annotator agreement? what were the rubrics for such annotation? This kind of information would help the community replicate or expand this subset, since it seems it is an important component of the method recipe.
>
> WildFeedback is an open-source dataset that already includes SAT/DSAT labels. For the seed subset, we collected conversations that naturally contain one dissatisfied response (DSAT) followed by a satisfied response (SAT). This corresponds to a common real-world interaction pattern in which the model revises its answer based on user feedback and ultimately receives a satisfactory rating. We curated this subset as seed data (491 samples) for warm-up training.
>
> > Q5: L263. There seems to be different reported percentages for 14B on AlpacaEval2, wrt to numbers in the abstract and the introduction (e.g. 12.11 vs 12.29 %)
>
> The abstract and introduction use the phrase **“up to”**, which refers to the **maximum** improvement observed across both WildFeedback and UltraFeedback settings.
>
> - On WildFeedback–14B, DRIFT achieves **+12.11%** AlpacaEval2 gain.
> - On UltraFeedback–14B, DRIFT achieves **+12.29%** AlpacaEval2 gain.
>
> Thus, the “up to 12.29%” in the abstract refers to the **best** 14B gain across all experiments.
>
> > W1 & Q9: Diversity-reward coverage analysis clarification
> > W1. The experiments section could benefit from more clarity and details, especially in the diversity-reward coverage analysis Q9: Section 4.3, L323. Could you please elaborate on how the embeddings were calculated? and how is the response reward calculated?
>
> We are happy to clarify:
> - **Embeddings.**
>     We use Qwen/Qwen3-Embedding-0.6B to compute embeddings for each response. These are then projected to 2D via UMAP.
> - **Reward.**
>     Each response is scored by the same reward model used elsewhere in the paper (OpenAssistant/reward-model-deberta-v3-large-v2).
>
> For a fixed prompt, we proceed as follows:
> 1. We collect 128 responses from each compared methods for that prompt (base model, DRIFT, SPIN, IterDPO), each with its associated scalar reward. This union of responses is what we refer to as “all responses” (total is 4x128 =512 responses).
> 2. We embed these responses into a semantic space and project them to 2D via UMAP.
> 3. On this shared 2D grid, we perform a reward-weighted Gaussian KDE over the union of all responses to obtain a global surface $Z_{\text {all }}(g) \in [0,1]$. This captures the overall high-reward topology of the response space.
> 4. The global high-reward region $\mathcal{H}$ is then defined as the set of grid points whose $Z_{\text {all}}(g)$ value is above the 0.8 quantile. For each method $m$, we compute its own reward-weighted surface $D_m(g)$ on the same grid and bandwidth, and measure what fraction of $H$ is covered by the regions where $D_m(g)$ is also above the same threshold.
>
> > Q10: Figure 3. Could you please elaborate on what the axes are, and their scales, of the UMAP plot on the left? Example responses from across the region would help us understand the semantic coverage of the represented region
>
> - **Axes meaning** : By definition, UMAP constructs a low-dimensional embedding by optimizing a graph-based approximation of the high-dimensional manifold. The resulting axes are simply coordinates of this learned manifold embedding，they do **not** correspond to interpretable semantic dimensions. Their sole role is to preserve **relative pairwise distances** induced by the original high-dimensional space.
>
> - **Figure layout**: The left panel shows only the embedded response points (colored by method), while the right panel overlays the reward-weighted KDE (“reward terrain”) on the same 2D grid. We keep both to separately visualize semantic spread and high-reward contours.
>
> - **Examples**: We already provide example responses for representative regions in the Figure 3 and  Appendix E.

---

> > ### Comment · Reviewer_aMHA · 2025-11-24
> >
> > Thanks for the clarifications.
> >
> > > For the seed subset, we collected conversations...
> >
> > Were the instances selected at random from WildFeedback or following specific criteria besides the user expressing flagging dissatisfaction?
> >
> >
> > > Axes meaning..the resulting axes are simply coordinates of this learned manifold embedding，they do not correspond to interpretable semantic dimensions...
> >
> > I see, the it would be better to clarify the sentence in L359 that makes reference to "semantic islands", which leads to a semantic interpretation of the space in Figure 3.

---

> > > ### Author Response · Authors · 2025-11-25
> > >
> > > We are happy to clarify this and make our paper easier for readers to understand!
> > >
> > > > For the seed subset, we collected conversations...
> > > > Q: Were the instances selected at random from WildFeedback or following specific criteria besides the user expressing flagging dissatisfaction?
> > >
> > > There was **no additional filtering or manual curation** beyond using user-provided feedback.
> > > The subset was **not randomly sampled** because we exhaustively scanned all WildFeedback conversations and extracted the cases where the user feedback transitioned from **Dissatisfaction (DSAT)** to **Satisfaction (SAT)**. And only **491** conversations in the dataset exhibited this DSAT→SAT pattern.
> > >
> > > The statistics of data are summarized below:
> > >
> > > | Data Category                     | # Conversations |
> > > | --------------------------------- | --------------- |
> > > | Conversations with DSAT           | **10,467**      |
> > > | Conversations with SAT            | **4,378**       |
> > > | Conversations with DSAT → SAT | **491**         |
> > >
> > > > Axes meaning..the resulting axes are simply coordinates of this learned manifold embedding，they do not correspond to interpretable semantic dimensions...
> > > > Q: I see, the it would be better to clarify the sentence in L359 that makes reference to "semantic islands", which leads to a semantic interpretation of the space in Figure 3.
> > >
> > > Thanks for your careful read and feedback! This analogy of “islands” refers to separated high-reward regions in the reward terrain figure. We aim to convey that DRIFT covers more separate high-reward regions, which correspond to diverse high-reward solutions for a given question.
> > >
> > > We will polish the writing here for better clarity, and we thank the reviewer again for the detailed and thoughtful feedback that helps us make the paper better.

---

> ### Author Response · Authors · 2025-11-20
> **Typo and Writing (Part 3/3)**
>
> ### **Typo and Writing**:
> > Q1: Eq2, L184: it is unclear what "l(*)" refers to since it does not appear in the equation
>
> This is a typo from an earlier draft. We will correct the equation and ensure all notation is consistent.
>
> > Q2: Algorithm 1 (L186) is never referenced in the text
>
> The algorithm is referenced earlier in the text (around L176), where we describe the procedure and point readers to the algorithm block. However, we agree that the current phrasing may not make this cross-reference sufficiently explicit and we will adjust our writing.
>
> > Q4: L243. The actual HF ID name would be better mentioned as a footnote or in the Appendix. In the main text, a short explanation of what this reward model is would be better for the story.
>
> Thanks, we will move to footnote.
>
> > Q8: Table 3. it seems that the wrong cell was bolded in the AlpacaEval2--14B-LC column and AlpacaEval2--7B-LC column. Please add an explanation in the caption.
>
> Appreciate your careful reading, we will fix this typo.

---

### Official Review · Reviewer_ukXJ · 2025-10-31

**Soundness:** 3
**Presentation:** 3
**Contribution:** 3
**Rating:** 6
**Confidence:** 3

**Summary:**

My understanding of the key contribution of the paper, is that performing iterative DPO while using an off-policy sample with negative feedback as the rejected response, and an on-policy generation from the previous model iteration as the chosen response (called DRIFT hereafter), produces better results than:
- The inverse i.e. iterative optimization where an off-policy sample with positive feedback is used as the chosen response, and an on-policy generation from the previous model iteration is used as the rejected response (called SPIN hereafter)
- On-policy self-play, where in each iteration multiple on-policy responses are generated and the model itself is used to choose the best and worst response as the chosen resp. rejected response (called IterDPO hereafter)

The “goodness” is measured via datasets UltraFeedback and WildFeedback, using WildBench (Elo) and AlpacaEval2 (Win rate).

This finding is relevant because off-policy samples with negative feedback are more ubiquitous than samples with positive feedback, in a context of real-world preference learning, i.e. beyond learning from human-annotated datasets, which are costly to produce.

**Strengths:**

- The rationale for the proposed approach and experiments is well explained, i.e., the need to move beyond annotation, followed by the observation that in a real-world setting negative feedback is more abundant than positive feedback.

- The experiments are well set up, including reruns based on the papers this paper builds on, i.e. a setup for SPIN, and a setup for IterDPO.

- The paper features valuable analyses, both empirical and theoretical.

**Weaknesses:**

- In the results we see that SPIN lags behind IterDPO, and IterDPO lags behind DRIFT. However, when looking at how the training samples are created in section 4.1 (Training Recipe), we see that for both IterDPO and DRIFT, in order to create the positive response, the model is guided, i.e. it is given the negative response and an instruction to improve. There does not seem to be an equivalent for SPIN: there is no instruction to create a “worse” negative relative to a reference point. I’m wondering to what extent this has an impact on the results.

**Questions:**

I have one main question, related to the weakness put forward above:  would it be feasible to rerun (part of) the experiment without giving InterDPO and DRIFT the reference point and an improvement instruction, so that we get an “apples to apples” comparison with SPIN, where the on-policy generation is not guided?
Please provide further motivation for the experimental choices, and if my question is deemed justified, additional insights from a more direct comparison may convince me to upgrade my score.

---

> ### Author Response · Authors · 2025-11-20
>
> Thank you for your positive feedback! We are happy to address your main concern from following two aspects:
>
> **1. New Unguided experiment**
> We re-trained DRIFT and iterDPO under an **Unguided** setting as you suggested, where generations do _not_ see a reference response or any explicit improvement instruction.
>
> - **DRIFT (Unguided).**
>     We remove the rejected response, user feedback, and improvement instruction entirely. The “chosen” response is now generated **only** from the original conversation prompt (exactly mirroring SPIN). The negative remains the off-policy DSAT response, consistent with DRIFT's asymmetric design.
>
> - **IterDPO (Unguided).**
>     IterDPO must generate both chosen and rejected responses on-policy. In the Unguided version, both responses are sampled from the same original prompt, and we use a reward model (OpenAssistant/reward-model-deberta-v3-large-v2) to rank them; the higher-scoring one becomes the chosen response. This introduces additional reward-model information in favor of IterDPO.
>
> We compare these Unguided variants to the original “full” version and to SPIN. The table below summarizes the results on WildBench and AlpacaEval2 :
>
> Table 1: Unguided Experiment Evaluation Results
>
> | Method      | Iter             | WildBench Score | Alpaca WinRate |
> | ----------- | ---------------- | --------------- | -------------- |
> | **SPIN**    | iter1            | 42.86           | 26.21          |
> |             | iter2            | 37.86           | 20.56          |
> | **IterDPO** | iter1 (Unguided) | 49.44           | 40.30          |
> |             | iter2 (Unguided) | 49.66           | 41.24          |
> |             | iter1 (full)     | 46.31           | 40.36          |
> |             | iter2 (full)     | 46.17           | 35.76          |
> | **DRIFT**   | iter1 (Unguided) | 50.77           | **46.77**      |
> |             | iter2 (Unguided) | 51.04           | 45.47          |
> |             | iter1 (full)     | 50.61           | 43.90          |
> |             | iter2 (full)     | **51.69**       | 46.64          |
> From this experiment, we could observe:
> - **DRIFT remains clearly stronger than SPIN and IterDPO even in the Unguided setting.**
> - **DRIFT (Unguided) ≈ DRIFT (full)** the performance difference is minimal.
>     This strongly suggests that DRIFT’s advantage does _not_ come mainly from the guidance prompt, but from its core design by anchoring on real DSAT negatives and samples positives dynamically from the evolving policy.
>
> **2. UltraFeedback (Sec 4.2.2) is a direct “apples to apples” comparison**
>
> We also want to clarify that the UltraFeedback experiment in the paper is inherently a controlled comparison where all methods use **the same prompts**, and only the positive/negative assignment mechanism differs. Specifically, for each UltraFeedback prompt:
>
> - **DRIFT:** sample a new chosen response from the prompt; use the original rejected response as the negative.
> - **IterDPO:** sample 5 on-policy responses from the prompt; use the reward model to pick best vs worst.
> - **SPIN:** use the original chosen response as the positive; sample a new rejected response from the prompt.
>
> Thus, all three methods share the same input prompt, same model family, and same training recipe. Table 3 in the paper (repeated below) shows that under this unified “apples to apples” setting, DRIFT consistently outperforms both SPIN and IterDPO at 7B and 14B:
>
> Table 2: Results of Training on UltraFeedback.
>
> | Method      | Iter  | WB Elo (7B) | WB Score (7B) | WB Elo (14B) | WB Score (14B) | Alpaca Win (7B) | Alpaca LC (7B) | Alpaca Win (14B) | Alpaca LC (14B) |
> | ----------- | ----- | ----------- | ------------- | ------------ | -------------- | --------------- | -------------- | ---------------- | --------------- |
> | Base        | –     | 1194.67     | 48.66         | 1213.17      | 55.08          | 37.69           | 39.73          | 36.65            | 43.58           |
> | **SPIN**    | iter1 | 1163.16     | 35.10         | 1178.88      | 36.99          | 18.39           | 26.62          | 16.09            | 28.20           |
> |             | iter2 | 1139.66     | 25.93         | 1155.07      | 28.01          | 13.23           | 19.91          | 13.66            | 24.29           |
> | **IterDPO** | iter1 | 1194.45     | 48.77         | 1214.14      | 54.21          | 34.53           | 40.55          | 33.29            | 42.84           |
> |             | iter2 | 1192.01     | 48.49         | 1215.12      | 54.78          | 32.15           | 39.92          | 28.51            | 40.47           |
> | **DRIFT**   | iter1 | 1197.04     | **50.91**     | 1215.67      | 58.52          | **41.04**       | 40.37          | 47.89            | **48.46**       |
> |             | iter2 | **1197.94** | 50.32         | **1218.75**  | **59.27**      | 40.47           | 37.09          | **48.94**        | 47.43           |
>
> We will clarify experiments and include the new Unguided ablation in the revised version.

---

> > ### Comment · Reviewer_ukXJ · 2025-11-25
> >
> > We thank the authors for taking the time to rerun the experiments we requested. The updated results are clear and meaningfully strengthen the contribution of the paper. We appreciate the additional effort, and based on these new findings we have increased our evaluation score.
> >
> > One point that would further enhance the revision is a short discussion of the underlying intuition behind the observed asymmetry — namely, why sampling the positive with an off-policy negative yields better results than sampling a negative with an off-policy positive. Even a concise explanation or hypothesis for this directional effect would help readers understand the mechanism driving the improvement.

---

> > > ### Author Response · Authors · 2025-11-25
> > >
> > > Thank you for the thoughtful follow-up and for increasing the evaluation score, we truly appreciate your careful review and the effort you put into helping us improve the paper.
> > >
> > > We will add a short discussion on the intuition behind the observed asymmetry. Briefly, our hypothesis is that sampling a fresh positive while keeping the negative off-policy preserves a cleaner and more informative margin: the new positive better reflects the current policy’s strengths, while the older negative still captures past failure modes. In contrast, pairing an off-policy positive with an on-policy negative often collapses this margin, as the stale positive no longer aligns with the model’s evolving distribution and the two responses become overly similar. We will include this explanation (and its limitations) in the revision to help readers better understand the mechanism behind the directional effect.
> > >
> > > Thank you again for the valuable feedback and it has meaningfully improved the clarity and depth of the paper.

---

### Official Review · Reviewer_i1Ue · 2025-11-01

**Soundness:** 3
**Presentation:** 3
**Contribution:** 2
**Rating:** 6
**Confidence:** 3

**Summary:**

This paper proposes DRIFT, a preference-optimization framework that leverages abundant real-world dissatisfaction (DSAT) signals rather than relying primarily on scarce explicit satisfaction (SAT) labels. DRIFT anchors learning on negative examples and dynamically samples positives from the model itself, promoting continued policy improvement. The authors provide theoretical guarantees showing non-vanishing gradients and expected utility improvements, and empirically demonstrate strong performance over IterDPO and SPIN across WildFeedback, WildBench, and AlpacaEval2 benchmarks.

**Strengths:**

- The paper provides a clear and practical motivation for leveraging DSAT signals, which are abundant in real-world user logs and therefore highly scalable compared to SAT-based approaches.

- The empirical results are strong, showing consistent improvements over iterative baselines across multiple benchmarks.

- The theoretical analysis is meaningful, demonstrating that DRIFT maintains a non-vanishing learning signal and guarantees improvement under local correlation assumptions with small update steps.

- The method is simple and scalable, integrating easily with existing preference-optimization pipelines without significant architectural changes.

**Weaknesses:**

- Baseline behavior appears inconsistent with prior literature: IterDPO and SPIN degrade after iteration here, whereas their original works report improvement. This raises questions about reproduction fidelity and tuning fairness.

- The method's reliance on self-generated positives may amplify noise or bias, potentially causing training instability, especially in iterative settings.

- Limited discussion on long-horizon stability; only up to iter-2 results are shown, making it unclear whether performance remains stable or oscillates over extended iterative training.

**Questions:**

- Theoretically, DRIFT maintains a gradient with a lower bound, but could this persistent gradient signal introduce instability?

- Have you evaluated more than two iterations? What happens when DRIFT is run for longer iterative cycles?

- I do not fully understand how Z_all (g) is computed in line 323. What exactly does "all responses" refer to in this context?

---

> ### Author Response · Authors · 2025-11-20
> **(Part 1/2)**
>
> We thank the reviewer for the thoughtful comments and for highlighting the strengths of our approach. Below we address the weaknesses and questions in detail.
>
> > **W1: Baseline behavior appears inconsistent with prior literature.**
>
> We believe that the degradation of SPIN and IterDPO in our results is **not** due to implementation issues, but arises from the **different setting** compared to their original papers.
>
> 1. Evaluating SPIN’s _official_ released models
> To directly address concerns about reproduction fidelity, we evaluated the SPIN models publicly released by the original authors ([https://huggingface.co/collections/UCLA-AGI/zephyr-7b-sft-full-spin](https://huggingface.co/collections/UCLA-AGI/zephyr-7b-sft-full-spin)) on WildBench.
>
> These models show:
> - a single-step improvement at iter1,
> - followed by consistent degradation from iter1 to iter3.
>
> Table 1: Official SPIN Models WildBench Task Scores
>
> | Model      | WB_Elo  | task_macro | Creative | Planning & Reasoning | Math & Data | Info/Advice | Coding | Length  |
> | ---------- | ------- | ---------- | -------- | -------------------- | ----------- | ----------- | ------ | ------- |
> | SPIN-iter0 | 1119.68 | 5.71       | 18.50    | 9.84                 | -4.21       | 18.07       | -3.02  | 1862.49 |
> | SPIN-iter1 | 1122.97 | 7.54       | 27.96    | 12.81                | -2.39       | 20.99       | -6.04  | 2326.51 |
> | SPIN-iter2 | 1121.30 | 7.39       | 24.38    | 12.71                | -1.28       | 19.13       | -4.85  | 2439.30 |
> | SPIN-iter3 | 1121.85 | 6.92       | 24.65    | 12.94                | -4.52       | 20.94       | -5.47  | 2782.45 |
>
> This matches our observation and suggests that **SPIN is well-suited for abundant, high-quality positive references but less robust in real-world DSAT-dominant regimes**, which directly motivates DRIFT.
>
> 2.  Different setting from original SPIN / IterDPO papers and controlled fair comparison
> SPIN was primarily evaluated on supervised SFT-style data with abundant high-quality positives, and IterDPO was mainly tested on synthetic or reward-model-labeled preferences. In contrast, our main results are on WildFeedback, which is extracted from noisy real-world user interactions, and we evaluate on WildBench and AlpacaEval2, which are **harder, long-form, multi-domain real-world tasks**. In this regime, we consistently observe that SPIN and IterDPO improve less, or even degrade in this more complex real-world setting, while DRIFT continues to improve.
>
> We implemented SPIN and IterDPO following their formulations and kept strictly controlled comparison by
> using the same model, the same training recipe (warm-start, optimizer, learning rate, β, sequence length, batch size, and number of epochs), the same WildFeedback or UltraFeedback splits as DRIFT for iterative DPO training. And for data generation, we keep the prompt for generating chosen same for DRIFT and IterDPO, the prompt for rejected response same for IterDPO and SPIN.
> Taken together, we believe these settings ensure a **fair and well-controlled experimental comparison.**
>
> > **W2 & Q1 Training Instability by self-generated positives and persistent gradient signal**
>
> 1. We understand your concern that self generated positives may amplify bias through iterations but our positives are **constrained by DSAT context as reference point and explicit improvement prompts**. This makes the self-generated positives targeted corrections of a known failure mode rather than arbitrary exploration, which reduces the chance of drifting into noise or biased behavior.
>
> 2. We also provide training dynamics in **Appendix D3** which shows stable improvement. And if DPO training collapse by noise or bias as reviewer concerned, we could easily diagnose the pattern in the training dynamics. However, in Figure 4, the loss curves exhibit stable convergence across iterations and the reward signals show the expected separation pattern: chosen rewards consistently increase while rejected rewards decrease. This trend is observed in both iterations, confirming the effectiveness of the training.
>
> 3. This question is also related to **longer iterative cycles (W3 & Q2)** and whether the method remains stable over more iterations. In our responses to W3 and Q2, we further show that DRIFT exhibits substantially greater stability than other methods in longer iterations, providing additional evidence that our approach prevents collapse from self-generated positives.

---

> ### Author Response · Authors · 2025-11-20
> **(Part 2/2)**
>
> > **W3 & Q2 Long-horizon performance (beyond 2 iterations).**
>
> We conducted additional experiments with **iter1 → iter5** on Qwen2.5-7B.
> Results are provided below:
>
> Table 2: WildBench Task Scores Across Five Iterations. Bold is the highest score for each methods.
>
> |Method|iter1|iter2|iter3|iter4|iter5|
> |---|---|---|---|---|---|
> |SPIN|**42.86**|37.86|33.76|34.62|33.99|
> |IterDPO|**46.31**|46.17|44.32|42.92|42.48|
> |**DRIFT**|50.61|51.69|51.36|**52.47**|51.22|
>
> We could observe that:
> 1. SPIN and IterDPO peak early and then degrade.
> 2. **DRIFT improves up to iter4**, then forms a stable plateau with minimal variation.
> 3. Limited collapse is observed even at iter5 for DRIFT.
>
> > **Q3: Clarify how Z_all (g) is computed and what"all responses" refer to**
>
> We are happy to clarify the construction of $Z_{\text {all }}(g)$ for you:
>
> For a fixed prompt, we proceed as follows:
>
> 1. We collect 128 responses from each compared methods for that prompt (base model, DRIFT, SPIN, IterDPO), each with its associated scalar reward. This union of responses is what we refer to as “all responses” (total is 4x128 =512 responses).
> 2. We embed these responses into a semantic space and project them to 2D via UMAP.
> 3. On this shared 2D grid, we perform a reward-weighted Gaussian KDE over the union of all responses to obtain a global surface $Z_{\text {all }}(g) \in [0,1]$. This captures the overall high-reward topology of the response space.
> 4. The global high-reward region $\mathcal{H}$ is then defined as the set of grid points whose $Z_{\text {all }}(g)$ value is above the 0.8 quantile. For each method $m$, we compute its own reward-weighted surface $D_m(g)$ on the same grid and bandwidth, and measure what fraction of $H$ is covered by the regions where $D_m(g)$ is also above the same threshold.

---

### Author Response · Authors · 2025-12-01
**Meta-summary for AC/PC (Part 2/2)**

### **3. Main Reviewer Concerns**
### **(1) Long-horizon stability beyond 2 iterations (Reviewer i1Ue, aMHA)**

The reviewers asked about the performance and stability for more iterations.

**Response: Added 5-iteration experiment during discussion**

We extended all methods to **iter1 → iter5** on Qwen2.5-7B, using an identical training recipe.

Table 1: WildBench Task Scores Across Five Iterations. Bold is the highest score for each methods.

|Method|iter1|iter2|iter3|iter4|iter5|
|---|---|---|---|---|---|
|SPIN|**42.86**|37.86|33.76|34.62|33.99|
|IterDPO|**46.31**|46.17|44.32|42.92|42.48|
|**DRIFT**|50.61|51.69|51.36|**52.47**|51.22|

We could observe that:
1. SPIN and IterDPO peak early and then degrade.
2. **DRIFT improves up to iter4**, then forms a stable plateau with minimal variation. Limited collapse is observed even at iter5 for DRIFT.

**Outcome (Reviewer reactions)**:
- This directly resolved Reviewer i1Ue, aMHA major concerns about DRIFT's performance and stability for longer iterative cycles.
- Reviewer aMHA acknowledged our method positively, noting that the extended iterative results offer **“a pretty illustrative showcase of the behaviour of DRIFT, and that it is performing as intended.”** Reviewer i1Ue did not have the opportunity to respond before the platform issue occurred.

### **(2) Whether DRIFT benefits unfairly from “guided” positive generation (Reviewer ukXJ)**

The reviewer questioned whether the **improvement instruction** and **DSAT reference** give DRIFT (and IterDPO) extra information compared to SPIN.

**Response: Unguided ablation conducted during discussion**

We implemented **unguided DRIFT** and **unguided IterDPO**, removing:
- reference DSAT response
- explicit improvement instruction
All methods then only saw the **original user prompt**, identical to SPIN.

We compare these Unguided variants to the original “full” version and to SPIN. The table below summarizes the results on WildBench and AlpacaEval2 :

Table 2: Unguided Experiment Evaluation Results

| Method      | Iter             | WildBench Score | Alpaca WinRate |
| ----------- | ---------------- | --------------- | -------------- |
| **SPIN**    | iter1            | 42.86           | 26.21          |
|             | iter2            | 37.86           | 20.56          |
| **IterDPO** | iter1 (Unguided) | 49.44           | 40.30          |
|             | iter2 (Unguided) | 49.66           | 41.24          |
|             | iter1 (full)     | 46.31           | 40.36          |
|             | iter2 (full)     | 46.17           | 35.76          |
| **DRIFT**   | iter1 (Unguided) | 50.77           | **46.77**      |
|             | iter2 (Unguided) | 51.04           | 45.47          |
|             | iter1 (full)     | 50.61           | 43.90          |
|             | iter2 (full)     | **51.69**       | 46.64          |

From this experiment, we could observe:
- **DRIFT remains clearly stronger than SPIN and IterDPO even in the Unguided setting.**
- **DRIFT (Unguided) ≈ DRIFT (full)** the performance difference is minimal.
This strongly suggests that DRIFT’s advantage does _not_ come mainly from the guidance prompt, but from its core design by anchoring on real DSAT negatives and samples positives dynamically from the evolving policy.

**Outcome (Reviewer reactions):**
- Reviewer ukXJ expressed satisfaction with this additional experiments and **increased score from 6 to 8**. "The updated results are clear and meaningfully strengthen the contribution of the paper. We appreciate the additional effort, and based on these new findings we have increased our evaluation score."

### **(3) Generalization Beyond One Model Family (Reviewer zGYz)**

The reviewer asked whether DRIFT’s effectiveness generalizes on different model families

**Response: Added Experiments on Gemma-3-12B-it (Different Architecture, Multimodal)**

During the discussion, we extended DRIFT to **Gemma-3-12B-it**, which differs from Qwen in both architecture and modality support. All methods (DRIFT, SPIN, IterDPO) were trained using the same recipe and evaluated on WildBench.Results on WildBench are shown below:

Table 3: Experiment Results on Gemma-3-12B-it

|Method|Iter|Score|Creative|Planning & Reasoning|Math & Data|Info/Advice|Coding|
|---|---|---|---|---|---|---|---|
|**Base**|—|60.77|66.10|62.46|54.10|65.21|59.62|
|**SPIN**|iter1|59.06|65.79|61.14|50.40|63.96|58.30|
||iter2|51.86|58.76|53.81|42.06|59.60|50.33|
|**IterDPO**|iter1|60.10|62.74|60.48|54.92|61.29|**62.10**|
||iter2|48.98|52.40|49.27|43.89|47.61|52.23|
|**DRIFT**|iter1|**61.73**|**66.37**|**63.02**|**56.19**|64.71|61.23|
||iter2|60.88|66.10|62.07|54.76|62.82|61.33|

These results confirm that DRIFT generalizes beyond a single model family and performs well even on a multimodal architecture.

**Outcome (Reviewer reactions):**
- This should resolve Reviewer zGYz’s major concern regarding whether DRIFT generalizes beyond a single model family.

---

### Author Response · Authors · 2025-12-01
**Meta summary for AC/PC (Part 1/2)**

We thank the reviewers for the constructive discussion. Below is a concise summary of the paper’s contributions, followed by the the most important concerns raised during review and how they were addressed, including additional experiments conducted during the discussion period.

### **1. Paper Contributions**

**DRIFT** introduces a scalable preference-learning framework built for real-world LLM deployments where **DSAT (dissatisfaction) signals are abundant** and SAT signals are sparse. Our contributions include:

- **(1) New paradigm for real-world preference learning**: DRIFT anchors learning on off-policy DSAT negatives and dynamically samples positives from the evolving policy. This directly leverage abundant natural user feedback (dissatisfaction) and requires no additional human labeling.
- **(2) Consistent strong empirical gains across datasets, scales, and model families:** Up to **+12.29% AlpacaEval2 win rate** and **+7.61% WildBench Score**, outperforming SPIN and IterDPO across WildFeedback and UltraFeedback.
- **(3) Long-horizon stability (after rebuttal):** DRIFT remains stable up to **five iterations**, forming a clear plateau without collapse.
- **(4) Diversity preservation:** Our reward-terrain analysis shows DRIFT preserves **higher diversity** and **broader high-reward coverage** than baseline methods, preventing the collapse into narrow response modes frequently seen in self-play or iterative self-improvement frameworks.
- **(5) Theoretical guarantees:** We prove that DRIFT **preserves preference margins** and **avoids gradient degeneration**, providing a principled explanation for its performance and stability.
- **(6) Additional safety analysis (after rebuttal):** We added **AdvBench** and **ToxiGen** evaluations. DRIFT shows **no increase** in jailbreak rate or toxicity, remaining aligned with baseline models.

### **2. Summary of Discussion Period**
The reviewer discussions and the accompanying additional experiments significantly strengthened the paper:
- **Longer-iteration** experiments demonstrated DRIFT’s **core stability and performance advantage**.
- More **model families** experiments.
- **Unguided ablations** addressed fairness concerns.
- Additional **safety analysis** for jailbreaking and toxicity.
- **Statistical significance** towards empirical gains
- Clarification / writing & typos as reviewer suggested

Reviewer ukXJ **raised scores (6 → 8)** after discussion, reviewer aMHA acknowledged improvements and give **positive feedback**. And two reviewers did not have the opportunity to respond **before the platform issue occurred.** We hope the AC/PC will consider this strengthened evidence and we sincerely appreciate all the time and effort invested in the review process for our community.

---

### Meta-Review · Area_Chair_nT8Z · 2025-12-31

**Summary:**

The decision was primarily informed by the authors' successful resolution of concerns regarding the stability and fairness of their iterative preference-learning framework. Initially, Reviewers i1Ue and aMHA expressed skepticism about whether DRIFT would suffer from model collapse or performance degradation over long training horizons, a common failure mode in self-improvement methods like SPIN and IterDPO. The authors addressed this by extending their experiments to five iterations, demonstrating that DRIFT maintains a stable performance plateau while baselines peak early and collapse. This longitudinal stability, coupled with a reward-terrain analysis showing that DRIFT preserves higher response diversity and broader high-reward coverage, convinced the reviewers that the method is robust for real-world iterative deployment.

A second pivotal concern, raised by Reviewer ukXJ, was whether DRIFT's superiority stemmed from an unfair advantage provided by the improvement instructions used during positive sample generation. To resolve this, the authors conducted an unguided ablation where all explicit guidance was removed. The results proved that even without these instructions, DRIFT significantly outperformed baselines, confirming that its core algorithmic design is the true driver of its success. Reviewer zGYz also noted initial concerns regarding cross-architecture generalization, which the authors mitigated by providing new, consistent results on the multimodal Gemma-3-12B-it model.

**Reviewer Concerns:**

The rebuttal process for DRIFT was highly effective, with the authors successfully resolving the vast majority of the reviewers' technical and empirical concerns. The most critical concerns addressed include:
* Long-Horizon Stability (i1Ue, aMHA): The authors provided a 5-iteration study showing DRIFT reaches a stable performance plateau, whereas SPIN and IterDPO degraded significantly after the first two iterations.
* Prompt-Guidance Bias (ukXJ): The authors performed a full-unguided ablation, proving that DRIFT's gains are an architectural benefit of anchoring on negatives rather than just a result of better prompting.
* Model Generalization (zGYz): The concern that the method only worked for Qwen2.5 was resolved by new experiments on Gemma-3-12B-it, demonstrating that the framework holds up on different architectures and multimodal models.
* Statistical and Safety Validity (aMHA, zGYz): The addition of bootstrap significance tests (confirming $p < 0.01$ for 14B models) and safety audits via AdvBench and ToxiGen effectively addressed the risks of iterative self-play and mode collapse.

There are virtually no significant outstanding concerns that would impede acceptance, though a few minor points remain as inherent characteristics of this research direction:
* Dependency on DSAT Quality: As noted by Reviewer zGYz, the method is still dependent on the accuracy of the dissatisfaction detector (SPUR).

**Reviewer Scores:**

Reviewer ukXJ raised scores from 6 to 8 after discussion.

The other reviewers are likely to raise their rating because I believe that the authors' rebuttals are very effective.

---

### Decision · Program_Chairs · 2026-01-26

Accept (Poster)